# Computational Guarantees for Doubly Entropic Wasserstein Barycenters via Damped Sinkhorn Iterations

**Tomas Vaškevičius and Lénaïc Chizat**
Institute of Mathematics
École Polytechnique Fédérale de Lausanne (EPFL) Station Z
CH-1015 Lausanne
Switzerland

## Abstract

We study the computation of doubly regularized Wasserstein barycenters, a recently introduced family of entropic barycenters governed by inner and outer regularization strengths. Previous research has demonstrated that various regularization parameter choices unify several notions of entropy-penalized barycenters while also revealing new ones, including a special case of debiased barycenters. In this paper, we propose and analyze an algorithm for computing doubly regularized Wasserstein barycenters. Our procedure builds on damped Sinkhorn iterations followed by exact maximization/minimization steps and guarantees convergence for any choice of regularization parameters. An inexact variant of our algorithm, implementable using approximate Monte Carlo sampling, offers the first non-asymptotic convergence guarantees for approximating Wasserstein barycenters between discrete point clouds in the free-support/grid-free setting.

## 1 Introduction

The Wasserstein distance between two probability distributions measures the least amount of effort needed to reconfigure one measure into the other. Unlike other notions of distances based solely on the numerical values taken by the distribution functions (e.g., the Kullback-Leibler divergence), the Wasserstein distance incorporates an additional layer of complexity by considering pairwise distances between distinct points, measured by some predetermined cost function. As a result, the Wasserstein distances can be seen to lift the geometry of the underlying space where the probability measures are defined to the space of the probability measures itself. This allows for a more thorough and geometrically nuanced understanding of the relationships between different probability measures, which proved to be a versatile tool of increasing importance in a broad spectrum of areas.

Given a collection of probability measures and an associated set of positive weights that sum to one, the corresponding Wasserstein barycenter minimizes the weighted sum of Wasserstein distances to the given measures. In the special case of two measures and the squared Euclidean cost function, Wasserstein barycenters concide with the notion of McCann's displacement interpolation introduced in the seminal paper [42]. The general case, encompassing an arbitrary number of measures, was first studied by Agueh and Carlier [1], where they also demonstrated a close link between Wasserstein barycenters and the multi-marginal optimal transport problem [29]. Recent years have witnessed an increasing number of applications of Wasserstein barycenters across various scientific disciplines. See, for instance, the following sample of works in economics [16, 11], statistics [8], image processing [50], and machine learning [19], among other areas. For further background and references we point the interested reader to the introductory surveys [47, 45] and the textbooks [58, 59, 54, 46, 28].

37th Conference on Neural Information Processing Systems (NeurIPS 2023).

Despite their compelling theoretical characteristics, the computation of Wasserstein barycenters poses significant computational challenges, particularly in large-scale applications. While Wasserstein barycenters can be computed in polynomial time for fixed dimensions [3], the approximation of Wasserstein barycenters is known to be NP–hard [4]. Currently employed methods for approximating Wasserstein barycenters are predominantly based on space discretizations. Unfortunately, such strategies are only computationally practical for problems of relatively modest scale. Although there are a handful of grid-free techniques available for approximating Wasserstein barycenters (e.g., [18, 35, 23, 40]), we are not aware of any existing methods that provide bounds on computational complexity. One contribution of the present paper is to introduce a method that in some regimes can provably approximate Wasserstein barycenters without relying on space discretizations, but instead employing approximate Monte Carlo sampling.

More broadly, the difficulties associated with computation of the optimal transport cost has prompted the exploration of computationally efficient alternatives, leading to the consideration of regularized Wasserstein distances. Among these, the entropic penalty has emerged as one of the most successful in applications. The practical success of entropic penalization can be attributed to Sinkhorn's algorithm [56], which enables efficient and highly parallelizable computation, an algorithm that gained substantial traction in the machine learning community following the work of Cuturi [20]. It is worth noting that entropic Wasserstein distances are of intrinsic interest, beyond their approximation capabilities. Indeed, they hold a rich historical connection to the Schrödinger bridge problem [55, 60, 26], as highlighted in the recent surveys [39, 14]. Furthermore, they increasingly serve as an analytically convenient tool for studying the unregularized optimal transport problem (see, e.g., [38, 31, 27, 15]) and they underlie some favorable statistical properties that are currently under active investigation; see the works [43, 30, 24, 48, 52, 49] and the references therein.

Let us now define the entropic optimal transport cost. Consider two probability measures, $\mu$ and $\nu$, both supported on $\mathcal{X}$, and let $c : \mathcal{X} \times \mathcal{X} \to [0, \infty)$ be a cost function. The entropic Wasserstein distance with a regularization level $\lambda > 0$ is defined as

$$T_\lambda(\mu, \nu) = \inf_{\gamma \in \Pi(\mu, \nu)} \mathbf{E}_{(X,Y) \sim \gamma}[c(X, Y)] + \lambda \mathrm{KL}(\gamma, \mu \otimes \nu), \tag{1}$$

where $\Pi(\mu, \nu)$ denotes the set of probability measures on $\mathcal{X} \times \mathcal{X}$ with marginal distributions equal to $\mu$ and $\nu$, and $\mathrm{KL}(\cdot, \cdot)$ is the Kullback-Leibler divergence. When $\lambda \to 0$, the regularized cost $T_\lambda(\mu, \nu)$ converges to the unregularized Wasserstein distance. Various properties of entropic optimal transport can be found in the recent lecture notes by Léonard [39].

To develop efficiently computable approximations for Wasserstein barycenters, a natural approach is to replace the unregularized Wasserstein cost with the minimizer of the weighted sum of entropy-regularized costs. This method was first explored by Cuturi and Doucet [21] and it has gained additional traction in the recent years. There is some flexibility in the definition of (1), which arises from substituting the reference product measure $\mu \otimes \nu$ with alternatives such as the Lebesgue measure. Consequently, various notions of entropic barycenters have emerged in the literature, which can be unified through the following optimization problem:

$$\min_\mu \sum_{j=1}^k w_j T_\lambda(\mu, \nu^j) + \tau \mathrm{KL}(\mu, \pi_{\mathrm{ref}}). \tag{2}$$

Here $\nu^1, \ldots, \nu^k$ are the probability measures whose barycenter we wish to compute and $w_1, \ldots, w_k$ are positive weights that sum to one. The inner regularization strength is denoted by $\lambda > 0$ while $\tau > 0$ is the outer regularization strength. The measure $\pi_{\mathrm{ref}}$ is an arbitrary reference measure, the support of which dictates the support of the computed barycenter. For instance, if we take $\pi_{\mathrm{ref}}$ to be a uniform measure on a particular discretization of the underlying space, we are dealing with a fixed-support setup. On the other hand, letting $\pi_{\mathrm{ref}}$ be the Lebesgue measure puts us in the free-support setup. We shall refer to the minimizer of (2) as the $(\lambda, \tau)$-barycenter, which exists and is unique due to the strict convexity of the outer regularization penalty; however, uniqueness may no longer holds when $\tau = 0$.

The objective (2) was recently studied in [17]; it also appeared earlier in [5] for the special case $\tau \geq \lambda$, where stochastic approximation algorithms were considered for the computation of fixed-support barycenters. In [17, Section 1.3], it is discussed how various choices of $(\lambda, \tau)$ relate to Barycenters previously explored in the literature. To provide a brief overview, $(0, 0)$ are the unregularized

Wasserstein barycenters studied in [1]. Inner-regularized barycenters $(\lambda, 0)$ introduce a shrinking bias; this can be seen already when $k = 1$, in which case the solution computes a maximum-likelihood deconvolution [53]. The $(\lambda, \lambda)$-barycenters were considered in [21, 7, 22, 10, 36]; they introduce a blurring bias. Likewise, blurring bias is introduced by the outer-regularized barycenters $(0, \tau)$, studied in [9, 12]. The only case not covered via the formulation (2) appears to be the one of debiased Sinkhorn barycenters [51, 33], for which an algorithm exists but without computational guarantees. Of particular interest are the $(\lambda, \lambda/2)$ barycenters: the choice $\tau = \lambda/2$ for smooth densities yields approximation bias of order $\lambda^2$, while the choice $\tau = \lambda$ results in bias of order $\lambda$, which is significantly larger than $\lambda^2$ in the regimes of interest. This is a new notion of entropic barycenters that was unveiled in the analysis of [17]. We provide the first convergence guarantees for this type of barycenters.

The regularity, stability, approximation, and statistical sample complexity properties of $(\lambda, \tau)$-barycenters were investigated in [17]. However, the question of obtaining non-asymptotic convergence guarantees for the computation of $(\lambda, \tau)$-barycenters with arbitrary regularization parameters was not addressed therein. In particular, the $(\lambda, \lambda/2)$ case, which has stood out due to its compelling mathematical features, has not yet been addressed in the existing literature. This gap is addressed by the present paper; we summarize our contributions in the following section.

## 1.1 Contributions

The remainder of this paper is organized as follows: Section 2 provides the necessary background on entropic optimal transport and a particular dual problem of the doubly regularized entropic objective (2). Section 3 introduces a damped Sinkhorn iteration scheme and complements it with convergence guarantees. An approximate version of the algorithm together with convergence results and implementation details is discussed in Section 4. We summarize our key contributions:

1. Lemma 1, presented in Section 3, demonstrates that bounds on the dual suboptimality gap for the dual problem (8), defined in Section 2.2, can be translated into Kullback-Leibler divergence bounds between the $(\lambda, \tau)$-barycenter and the barycenters corresponding to dual-feasible variables. This translation enables us to formulate all our subsequent results in terms of optimizing the dual objective (8).

2. In Section 3, we introduce a damped Sinkhorn scheme (Algorithm 1) that can be employed to optimize $(\lambda, \tau)$-barycenters for any choice of regularization parameters. The damping factor $\min(1, \tau/\lambda)$ accommodates the degrading smoothness properties of the dual objective (8) as a function of decreasing outer regularization parameter $\tau$. The introduced damping of the Sinkhorn iterations is, in fact, necessary and it is one of our core contributions: undamped exact scheme can be experimentally shown to diverge as soon as $\tau < \lambda/2$.

3. The main result of this paper is Theorem 1 proved in Section 3. It provides convergence guarantees for Algorithm 1 with arbitrary choice of regularization parameters $\lambda, \tau > 0$. This, in particular, results in the first algorithm with guarantees for computing $(\lambda, \lambda/2)$ barycenters. For smooth densities, these barycenters incur a bias of order $\lambda^2$ in contrast to the predominantly studied $(\lambda, \lambda)$ barycenters that incur bias of order $\lambda$.

4. In Section 4, we describe Algorithm 2, an extension of Algorithm 1 that allows us to perform inaccurate updates. We formulate sufficient conditions on the inexact updates oracle under which the errors in the convergence analysis do not accumulate. Section 4.1 details an implementation of this inexact oracle, based on approximate Monte Carlo sampling.

5. Theorem 2 proved in Section 4 furnishes convergence guarantees for Algorithm 2. When combined with the implementation of the inexact oracle described in Section 4.1, this yields a provably convergent scheme for a grid-free computation of entropic Wasserstein barycenters between discrete distributions, provided sufficient regularity on the domain $\mathcal{X}$ and the cost function $c$.

6. Appendix F complements our theoretical results with numerical experiments. Our simulations experimentally confirm the necessity of damping when $\tau < \lambda/2$. They also provide experimental support for the suggested damping factor in Algorithms 1 and 2.

## 2  Background and Notation

This section provides the background material on doubly regularized entropic Wasserstein barycenters and introduces the notation used throughout the paper. In the remainder of the paper, let $\mathcal{X}$ be a compact and convex subset of $\mathbb{R}^d$ with a non-empty interior. Let $\mathcal{P}(\mathcal{X})$ denote the set of probability measures on $\mathcal{X}$ endowed with Borel sigma-algebra. Let $c : \mathcal{X} \times \mathcal{X} \to [0, \infty)$ be a cost function such that $c_\infty(\mathcal{X}) = \sup_{x,x' \in \mathcal{X}} c(x, x') < \infty$. We denote by $\mathrm{KL}(\cdot, \cdot)$ the Kullback-Leibler divergence, $\|\cdot\|_{\mathrm{TV}}$ is the total-variation norm, and $\|f\|_{\mathrm{osc}} = \sup_x f(x) - \inf_{x'} f(x')$ is the oscillation norm. Given two measures $\nu, \nu'$, the notation $\nu \ll \nu'$ denotes that $\nu$ is absolutely continuous with respect to the measure $\nu'$; in this case $d\nu/d\nu'$ denotes the Radon-Nikodym derivative of $\nu$ with respect to $\nu'$. Finally, throughout the paper $w$ denotes a vector of $k$ strictly positive elements that sum to one.

### 2.1  Entropic Optimal Transport

For any $\mu, \nu \in \mathcal{P}(\mathcal{X})$ define the entropy regularized optimal transport problem by

$$T_\lambda(\mu, \nu) = \inf_{\gamma \in \Pi(\mu, \nu)} \mathbf{E}_{(X,Y) \sim \gamma}[c(X, Y)] + \lambda \mathrm{KL}(\gamma, \mu \otimes \nu), \tag{3}$$

where KL is the Kullback-Leibler divergence and $\Pi(\mu, \nu) \subseteq \mathcal{P}(\mathcal{X} \otimes \mathcal{X})$ is the set of probability measures such that for any $\gamma \in \Pi(\mu, \nu)$ and any Borel subset $A$ of $\mathcal{X}$ it holds that $\gamma(A \times \mathcal{X}) = \mu(A)$ and $\gamma(\mathcal{X} \times A) = \nu(A)$.

Let $E_\lambda^{\mu,\nu} : L_1(\mu) \times L_1(\nu) \to \mathbb{R}$ be the function defined by

$$E_\lambda^{\mu,\nu}(\phi, \psi) = \mathbf{E}_{X \sim \mu}[\phi(X)] + \mathbf{E}_{Y \sim \nu}[\psi(Y)]$$
$$+ \lambda \left( 1 - \int_\mathcal{X} \int_\mathcal{X} \exp\left( \frac{\phi(x) + \psi(y) - c(x,y)}{\lambda} \right) \nu(dy)\mu(dx) \right).$$

The entropic optimal transport problem (3) admits the following dual representation:

$$T_\lambda(\mu, \nu) = \max_{\phi, \psi} E_\lambda^{\mu,\nu}(\phi, \psi). \tag{4}$$

For any $\psi$ define

$$\phi_\psi \in \mathrm{argmax}_{\phi \in L_1(\mu)} E_\lambda^{\mu,\nu}(\phi, \psi).$$

The solution is unique $\mu$-almost everywhere up to a constant; we fix a particular choice

$$\phi_\psi(x) = -\lambda \log \left( \int_\mathcal{X} \exp\left( \frac{\psi(y) - c(x,y)}{\lambda} \right) \nu(dy) \right).$$

Likewise, we denote $\psi_\phi = \mathrm{argmax}_{\psi \in L_1(\nu)} E_\lambda^{\mu,\nu}(\phi, \psi)$ with the analogous expression to the one given above, interchanging the roles of $\phi$ and $\psi$. Then, the maximum in (4) is attained by any pair $(\phi^*, \psi^*)$ such that $\phi^* = \phi_{\psi^*}$ and $\psi^* = \psi_{\phi^*}$; such a pair is said to solve the Schrödinger system and it is unique up to translations $(\phi^* + a, \psi^* - a)$ by any constant $a \in \mathbb{R}$. The optimal coupling that solves the primal problem (3) can be obtained from the pair $(\phi^*, \psi^*)$ via the primal-dual relation

$$\gamma^*(dx, dy) = \exp\left( \frac{\phi^*(x) + \psi^*(y) - c(x,y)}{\lambda} \right) \mu(dx)\nu(dy).$$

We conclude this section by listing two properties of functions of the form $\phi_\psi$. These properties will be used repeatedly throughout this paper. First, for any $\psi$ we have

$$\int_\mathcal{X} \int_\mathcal{X} \exp\left( \frac{\phi_\psi(x) + \psi(y) - c(x,y)}{\lambda} \right) \nu(dy)\mu(dx) = 1,$$

which means, in particular, that for any $\psi$ we have

$$E_\lambda^{\mu,\nu}(\phi_\psi, \psi) = \mathbf{E}_{X \sim \mu}[\phi_\psi(X)] + \mathbf{E}_{Y \sim \nu}[\psi(Y)]. \tag{5}$$

The second property of interest is that for any $\psi$ and any $x, x' \in \mathcal{X}$ it holds that

$$\phi_\psi(x) - \phi_\psi(x') = -\lambda \log \frac{\int \exp\left(\frac{\psi(y)-c(x,y)}{\lambda}\right)\nu(dy)}{\int \exp\left(\frac{\psi(y)-c(x',y)}{\lambda}\right)\nu(dy)}$$

$$= -\lambda \log \frac{\int \exp\left(\frac{\psi(y)-c(x',y)+c(x',y)-c(x,y)}{\lambda}\right)\nu(dy)}{\int \exp\left(\frac{\psi(y)-c(x',y)}{\lambda}\right)\nu(dy)}$$

$$\leq \sup_{y\in\mathcal{X}} c(x',y) - c(x,y) \leq c_\infty(\mathcal{X}).$$

In particular, for any $\psi$ we have

$$\|\phi_\psi\|_{\mathrm{osc}} = \sup_x \phi_\psi(x) - \inf_{x'} \phi_\psi(x') \leq c_\infty(\mathcal{X}). \tag{6}$$

## 2.2  Doubly Regularized Entropic Barycenters

Let $\boldsymbol{\nu} = (\nu^1, \ldots, \nu^k) \in \mathcal{P}(\mathcal{X})^k$ be $k$ probability measures and let $w \in \mathbb{R}^k$ be a vector of positive numbers that sum to one. Given the inner regularization strength $\lambda > 0$ and the outer regularization strength $\tau > 0$, the $(\lambda, \tau)$ barycenter $\mu_{\lambda,\tau} \in \mathcal{P}(\mathcal{X})$ of probability measures $\boldsymbol{\nu}$ with respect to the weights vector $w$ is defined as the unique solution to the following optimization problem:

$$\mu_{\lambda,\tau} = \mathrm{argmin}_{\mu\in\mathcal{P}(\mathcal{X})} \sum_{j=1}^k w_j T_\lambda(\mu, \nu^j) + \tau \mathrm{KL}(\mu, \pi_{\mathrm{ref}}), \tag{7}$$

where $\pi_{\mathrm{ref}} \in \mathcal{P}(\mathcal{X})$ is a reference probability measure.

We will now describe how to obtain a concave dual maximization problem to the primal problem (7), following along the lines of Chizat [17, Section 2.3], where the interested reader will find a comprehensive justification of all the claims made in the rest of this section.

First, using the semi-dual formulation of entropic optimal transport problem (5), we have, for each $j \in \{1, \ldots, k\}$

$$T_\lambda(\mu, \nu^j) = \sup_{\psi^j \in L_1(\nu^j)} \mathbf{E}_{X\sim\mu}[\phi_{\psi^j}(X)] + \mathbf{E}_{Y\sim\nu^j}[\psi^j(Y)].$$

Denote $\boldsymbol{\psi} = (\psi^1, \ldots, \psi^j) \in L_1(\boldsymbol{\nu})$. Then, we may rewrite the primal problem (7) by

$$\min_{\mu\in\mathcal{P}(X)} \max_{\boldsymbol{\psi}\in L_1(\boldsymbol{\nu})} \sum_{j=1}^k w_j \mathbf{E}_{Y\sim\nu^j}\left[\psi^j(Y)\right] + \mathbf{E}_{X\sim\mu}\left[\sum_{j=1}^k w_j \phi_{\psi^j}(X)\right] + \tau \mathrm{KL}(\mu, \pi_{\mathrm{ref}}).$$

Interchanging $\min$ and $\max$, which is justified using compactness of $\mathcal{X}$ as detailed in [17], we obtain the dual optimization objective $E_{\lambda,\tau}^{\boldsymbol{\nu},w} : L_1(\boldsymbol{\nu}) \to \mathbb{R}$ defined by

$$E_{\lambda,\tau}^{\boldsymbol{\nu},w}(\boldsymbol{\psi}) = \min_{\mu\in\mathcal{P}(X)} \sum_{j=1}^k w_j \mathbf{E}_{Y\sim\nu^j}\left[\psi^j(Y)\right] + \mathbf{E}_{X\sim\mu}\left[\sum_{j=1}^k w_j \phi_{\psi^j}(X)\right] + \tau \mathrm{KL}(\mu, \pi_{\mathrm{ref}}).$$

$$= \sum_{j=1}^k w_j \mathbf{E}_{Y\sim\nu^j}\left[\psi^j(Y)\right] - \tau \log \int \exp\left(\frac{-\sum_{j=1}^k w_j \phi_{\psi^j}(x)}{\tau}\right)\pi_{\mathrm{ref}}(dx). \tag{8}$$

The infimum above is attained by the measure

$$\mu_{\boldsymbol{\psi}}(dx) = Z_{\boldsymbol{\psi}}^{-1} \exp\left(\frac{-\sum_{j=1}^k \phi_{\psi^j}(x)}{\tau}\right)\pi_{\mathrm{ref}}(dx), \quad Z_{\boldsymbol{\psi}} = \int \exp\left(\frac{-\sum_{j=1}^k \phi_{\psi^j}(x)}{\tau}\right)\pi_{\mathrm{ref}}(dx).$$

To each dual variable $\boldsymbol{\psi}$ we associate the marginal measures $\nu_{\boldsymbol{\psi}}^j(dy)$ defined for $j = 1, \ldots, k$ by

$$\nu_{\boldsymbol{\psi}}^j(dy) = \nu^j(dy) \int \exp\left(\frac{\phi_{\psi^j}(x) + \psi^j(y) - c(x,y)}{\lambda}\right)\mu_{\boldsymbol{\psi}}(dx). \tag{9}$$

Finally, we mention that the objective $E_{\lambda,\tau}^{\boldsymbol{\nu},w}$ is concave and for any $\boldsymbol{\psi}, \boldsymbol{\psi}'$ it holds that

$$\lim_{h \to 0} \frac{E_{\lambda,\tau}^{\boldsymbol{\nu},w}(\boldsymbol{\psi} + h\boldsymbol{\psi}') - E_{\lambda,\tau}^{\boldsymbol{\nu},w}(\boldsymbol{\psi})}{h} = \sum_{j=1}^{k} w_j \left( \mathbf{E}_{\nu^j}[(\psi')^j] - \mathbf{E}_{\nu_{\boldsymbol{\psi}}^j}[(\psi')^j] \right) .$$

In particular, fixing any optimal dual variable $\boldsymbol{\psi}^*$, for any $\boldsymbol{\psi}$ it holds using concavity of $E_{\lambda,\tau}^{\boldsymbol{\nu},w}$ that

$$0 \le E_{\lambda,\tau}^{\boldsymbol{\nu},w}(\boldsymbol{\psi}^*) - E_{\lambda,\tau}^{\boldsymbol{\nu},w}(\boldsymbol{\psi}) \le \sum_{j=1}^{k} w_k \left( \mathbf{E}_{\nu^j} \left[ (\psi^*)^j - \psi^j \right] - \mathbf{E}_{\nu_{\boldsymbol{\psi}}^j} \left[ (\psi^*)^j - \psi^j \right] \right). \tag{10}$$

This concludes our overview of the background material on $(\lambda, \tau)$-barycenters.

## 3 Damped Sinkhorn Scheme

This section introduces a damped Sinkhorn-based optimization scheme (Algorithm 1) and provides guarantees for its convergence (Theorem 1). Before describing the algorithm, we make a quick detour to the following lemma, proved in Appendix A, which shows that the sub-optimality gap bounds on the dual objective (8) can be transformed into corresponding bounds on relative entropy between the $(\lambda, \tau)$-barycenter and the barycenter associated to a given dual variable.

**Lemma 1.** *Fix any $\lambda, \tau > 0$ and $\boldsymbol{\nu}, w$. Let $\boldsymbol{\psi}^*$ be the maximizer of dual problem $E_{\lambda,\tau}^{\boldsymbol{\nu},w}$ and let $\mu_{\boldsymbol{\psi}^*}$ be the corresponding minimizer of the primal objective (7). Then, for any $\boldsymbol{\psi} \in L_1(\boldsymbol{\nu})$ we have*

$$\mathrm{KL}(\mu_{\boldsymbol{\psi}^*}, \mu_{\boldsymbol{\psi}}) \le \tau^{-1}(E_{\lambda,\tau}^{\boldsymbol{\nu},w}(\boldsymbol{\psi}^*) - E_{\lambda,\tau}^{\boldsymbol{\nu},w}(\boldsymbol{\psi})).$$

We now turn to describing an iterative scheme that ensures convergence of the dual suboptimality gap to zero. Let $\boldsymbol{\psi}_t$ be an iterate at time $t$. Then, we have

$$E_{\lambda,\tau}^{\boldsymbol{\nu},w}(\boldsymbol{\psi}_t) = L(\boldsymbol{\psi}_t, \boldsymbol{\phi}_t, \mu_t) = \sum_{j=1}^{k} w_j \mathbf{E}_{\nu^j}[\psi_t^j] - \mathbf{E}_{\mu_t}[\phi_t^j] + \tau\mathrm{KL}(\mu_t, \pi_{\mathrm{ref}}),$$

where

$$\phi^j = \mathrm{argmax}_\phi E_\lambda^{\mu_{t-1},\nu^j}(\phi, \psi_t^j) \quad \text{and} \quad \mu_t = \mathrm{argmin}_\mu \left\{ \mathbf{E}_\mu \Big[ \sum_j w_j \phi_t^j \Big] + \tau\mathrm{KL}(\mu, \pi_{\mathrm{ref}}) \right\}. \tag{11}$$

In particular, when optimizing the dual objective $E_{\lambda,\tau}^{\boldsymbol{\nu},w}$, every time the variable $\boldsymbol{\psi}_t$ is updated, it automatically triggers the exact maximization/minimization steps defined in (11). It is thus a natural strategy to fix $\boldsymbol{\phi}_t$ and $\mu_t$ and perform exact minimization on $\boldsymbol{\psi}$, which can be done in closed form:

$$\psi_{t+1}^j = \mathrm{argmax}_\psi E_\lambda^{\mu_t,\nu^j}(\phi_t^j, \psi) = \psi_t^j - \lambda \log \frac{d\nu_t^j}{d\nu^j}, \tag{12}$$

where $\nu_t^j$ denotes the marginal distribution $\nu_{\boldsymbol{\psi}_t}^j$ defined in (9). The update (12) performs a Sinkhorn update on each block of variables $\psi^j$. Together, the update (12) followed by (11) results in the iterative Bregman projections algorithm introduced in [7]. In [36], it was shown that this scheme converges for the $(\lambda, \lambda)$-barycenters. The analysis of [36] is built upon a different dual formulation from the one considered in our work; this alternative formulation is only available when $\tau = \lambda$ [17, Section 2.3] and thus excludes the consideration of debiased barycenters $(\lambda, \lambda/2)$.

We have observed empirically (see Appendix F) that the iterates of the iterative Bregman projections (i.e., the scheme of updates defined in (12) and (11)) diverge whenever $\tau < \lambda/2$. Indeed, decreasing the outer regularization parameter $\tau$ makes the minimization step in (11) less stable. As a result, the cumulative effect of performing the updates (12) and (11) may result in a decrease in the value of the optimization objective $E_{\lambda,\tau}^{\boldsymbol{\nu},w}$.

One of the main contributions of our work is to show that this bad behaviour can be mitigated by damping the exact Sinkhorn updates (12). This leads to Algorithm 1 for which convergence guarantees are provided in Theorem 1 stated below.

**Algorithm 1:** Exact Damped Sinkhorn Scheme

---

**Input:** regularization strengths $\lambda, \tau > 0$, reference measure $\pi_{\text{ref}}$, number of iterations $T$ and $k$ marginal measures $\nu^1, \ldots, \nu^k$ with positive weights $w_1, \ldots, w_k$ such that $\sum_{j=1}^{k} w_j = 1$.

1. Set $\eta = \min(1, \tau/\lambda)$ and initialize $(\psi_0^j) = 0$ for $j \in \{1, \ldots, k\}$.

2. For $t = 0, 1 \ldots, T - 1$ do

   (a) $\phi_t^j(x) \leftarrow -\lambda \log \int_{\mathcal{X}} \exp((\psi_t^j(y) - c(x, y))/\lambda)\nu^j(dy)$ for $j \in \{1, \ldots, k\}$

   (b) $V_t(x) \leftarrow \sum_{j=1}^{k} w_j \phi^j(t)(x)$

   (c) $Z_t \leftarrow \int \exp(-V_t(x)/\tau)d\pi_{\text{ref}}(dx)$

   (d) $\mu_t(dx) \leftarrow Z_t^{-1} \exp(-V_t(x)/\tau)\pi_{\text{ref}}(dx)$

   (e) $\frac{d\nu_t^j}{d\nu^j}(y) \leftarrow \int \exp\left(\frac{\phi_t^j(x) + \psi_t^j(y) - c(x,y)}{\lambda}\right)\mu_t(dx)$ for $j \in \{1, \ldots, k\}$.

   (f) $\psi_{t+1}^j(y) \leftarrow \psi_t^j(y) - \eta\lambda \log \frac{d\nu_t^j}{d\nu^j}(y)$ for $j \in \{1, \ldots, k\}$.

3. Return $(\phi_T^j, \psi_T^j)_{j=1}^k$.

---

**Theorem 1.** *Fix any $\lambda, \tau > 0$ and $\boldsymbol{\nu}, w$. Let $\psi^*$ be the maximizer of dual problem $E_{\lambda,\tau}^{\boldsymbol{\nu},w}$. Let $(\boldsymbol{\psi}_t)_{t \geq 0}$ be the sequence of iterates generated by Algorithm 1. Then, for any $t \geq 1$ it holds that*

$$E_{\lambda,\tau}^{\boldsymbol{\nu},w}(\boldsymbol{\psi}^*) - E_{\lambda,\tau}^{\boldsymbol{\nu},w}(\boldsymbol{\psi}_t) \leq \frac{2c_\infty(\mathcal{X})^2}{\min(\lambda, \tau)} \frac{1}{t}.$$

Our convergence analysis draws upon the existing analyses of Sinkhorn's algorithm [2, 25], which in turn are based on standard proof strategies in smooth convex optimization (e.g., [44, Theorem 2.1.14]). Concerning the proof of Theorem 1, the main technical contribution of our work lies in the following proposition proved in Appendix B.

**Proposition 1.** *Consider the setup of Theorem 1. Then, for any integer $t \geq 0$ it holds that*

$$E_{\lambda,\tau}^{\boldsymbol{\nu},w}(\boldsymbol{\psi}_{t+1}) - E_{\lambda,\tau}^{\boldsymbol{\nu},w}(\boldsymbol{\psi}_t) \geq \min(\tau, \lambda) \sum_{j=1}^{k} w_j \text{KL}(\nu^j, \nu_t^j).$$

With Proposition 1 at hand, we are ready to prove Theorem 1.

*Proof of Theorem 1.* Denote $\delta_t = E_{\lambda,\tau}^{\boldsymbol{\nu},w}(\boldsymbol{\psi}^*) - E_{\lambda,\tau}^{\boldsymbol{\nu},w}(\boldsymbol{\psi}_t)$. We would like to relate the suboptimality gap $\delta_t$ to the increment $\delta_t - \delta_{t+1}$. To do this, we will first show that the iterates $\boldsymbol{\psi}_t$ have their oscillation norm bounded uniformly in $t$. Indeed, for any $j \in \{1, \ldots, k\}$, any $t \geq 1$, and any $y \in \mathcal{X}$ we have

$$\psi_t^j(y) = (1 - \eta)\psi_{t-1}^j(y) + \eta\psi_{\phi_t^j}(y).$$

By (6), $\psi_{\phi_t^j}$ has oscillation norm bounded by $c_\infty(\mathcal{X})$. Because $\psi_0^j = 0$ and $\eta \in (0, 1]$, by induction on $t$ it follows that $\|\psi_t\|_{\text{osc}} \leq c_\infty(\mathcal{X})$ for any $t \geq 0$. Combining the bound on the dual sub-optimality gap (10) with Pinsker's inequality yields

$$\delta_t \leq 2c_\infty(\mathcal{X}) \sum_{j=1}^{k} w_j \|\nu^j - \nu_t^j\|_{\text{TV}} \leq \sqrt{2}c_\infty \sum_{j=1}^{k} w_j \sqrt{\text{KL}(\nu^j, \nu_t^j)}.$$

Using concavity of the square root function, Proposition 1 yields for any $t \geq 0$

$$\delta_t - \delta_{t+1} \geq \min(\lambda, \tau) \sum_{j=1}^{k} w_j \text{KL}(\nu^j, \nu_t^j) \geq \frac{\min(\lambda, \tau)}{2c_\infty(\mathcal{X})^2} \delta_t^2.$$

By Proposition 1, the sequence $\delta_t$ is non-increasing. Hence, dividing the above equality by $\delta_t \delta_{t+1}$ yields

$$\frac{1}{\delta_{t+1}} - \frac{1}{\delta_t} \geq \frac{\min(\lambda, \tau)}{2c_\infty(\mathcal{X})^2}.$$

Telescoping the left hand side completes the proof. $\qquad\square$

# 4 Approximate Damped Sinkhorn Scheme

In this section, we extend the analysis of Algorithm 1 to an approximate version of the algorithm. Then, in Section 4.1, we describe how inexact updates may be implemented via approximate random sampling, thus enabling the computation of $(\lambda, \tau)$-barycenters in the free-support setting with convergence guarantees.

Algorithm 2 describes an inexact version of Algorithm 1. It replaces the damped Sinkhorn iterations of Algorithm 1 via approximate updates computed by an approximate Sinkhorn oracle – a procedure that satisfies the properties listed in Definition 1.

**Definition 1** (Approximate Sinkhorn Oracle). An $\varepsilon$-approximate Sinkhorn oracle is a procedure that given any $\psi$ and any index $j \in \{1, \ldots, k\}$, returns a Radon-Nikodym derivative $\frac{d\widetilde{\nu}_{\psi}^j}{d\nu^j}$ of a measure $\widetilde{\nu}_{\psi}^j \ll \nu^j$ that satisfies the following properties:

1. $\frac{d\widetilde{\nu}_{\psi}^j}{d\nu^j}$ is strictly positive on the support of $\nu^j$;

2. $\|\widetilde{\nu}_{\psi}^j - \nu_{\psi}^j\|_{\mathrm{TV}} \leq \varepsilon/(2c_\infty(\mathcal{X}))$;

3. $\mathbf{E}_{Y \sim \nu^j}\big[\frac{d\nu_{\psi}^j}{d\widetilde{\nu}_{\psi}^j}(Y)\big] \leq 1 + \varepsilon^2/(2c_\infty(\mathcal{X})^2)$;

4. For any $\eta \in [0,1]$ and any $j \in \{1, \ldots, k\}$ it holds that $\|\psi^j + \eta\lambda \log(d\widetilde{\nu}_{\psi}^j/d\nu^j)\|_{\mathrm{osc}} \leq (1-\eta)\|\psi^j\|_{\mathrm{osc}} + \eta c_\infty(\mathcal{X})$.

---

**Algorithm 2:** Approximate Damped Sinkhorn Scheme

---

**Input:** error tolerance parameter $\varepsilon > 0$, a function "ApproximateSinkhornOracle" satisfying properties listed in Definition 1, regularization strengths $\lambda, \tau > 0$, reference measure $\pi_{\mathrm{ref}}$, number of iterations $T$ and $k$ marginal measures $\nu^1, \ldots, \nu^k$ with positive weights $w_1, \ldots, w_k$ such that $\sum_{j=1}^k w_j = 1$.

1. Set $\eta = \min(1, \tau/\lambda)$ and initialize $(\psi_0^j) = 0$ for $j \in \{1, \ldots, k\}$.
2. For $t = 0, 1 \ldots, T-1$ do
   (a) $\frac{d\widetilde{\nu}_t^j}{d\nu^j}(y) \leftarrow \mathrm{ApproximateSinkhornOracle}(\boldsymbol{\nu}, \lambda, \tau, \boldsymbol{\psi}_t, \varepsilon, j)$ for $j \in \{1, \ldots, k\}$.
   (b) $\psi_{t+1}^j(y) \leftarrow \psi_t^j(y) - \eta\lambda \log \frac{d\widetilde{\nu}_t^j}{d\nu^j}(y)$ for $j \in \{1, \ldots, k\}$.
3. Return $(\phi_T^j, \psi_T^j)_{j=1}^k$.

---

The following theorem shows that Algorithm 2 enjoys the same convergence guarantees as Algorithm 1 up to the error tolerance of the procedure used to implement the approximate updates. A noteworthy aspect of the below theorem is that the error does not accumulate over the iterations.

**Theorem 2.** *Fix any $\lambda, \tau > 0$ and $\boldsymbol{\nu}, w$. Let $\psi^*$ be the maximizer of dual problem $E_{\lambda,\tau}^{\boldsymbol{\nu},w}$. Let $(\widetilde{\psi}_t)_{t\geq 0}$ be the sequence of iterates generated by Algorithm 2 with the accuracy parameter $\varepsilon \geq 0$. Let $T = \min\{t : E_{\lambda,\tau}^{\boldsymbol{\nu},w}(\psi^*) - E_{\lambda,\tau}^{\boldsymbol{\nu},w}(\widetilde{\psi}_t) \leq 2\varepsilon\}$. Then, for any $t \leq T$ it holds that*

$$E_{\lambda,\tau}^{\boldsymbol{\nu},w}(\psi^*) - E_{\lambda,\tau}^{\boldsymbol{\nu},w}(\widetilde{\psi}_t) \leq 2\varepsilon + \frac{2c_\infty(\mathcal{X})^2}{\min(\lambda,\tau)}\frac{1}{t}.$$

The proof of the above theorem can be found in Appendix C.

## 4.1 Implementing the Approximate Sinkhorn Oracle

In this section, we show that the approximate Sinkhorn oracle (see Definition 1) can be implemented using approximate random sampling when the marginal distributions $\nu^j$ are discrete. To this end, fix

the regularization parameters $\lambda, \tau > 0$, the weight vector $w$, and consider a set of $k$ discrete marginal distributions

$$\nu^j = \sum_{l=1}^{m_j} \nu^j(y_l^j)\delta_{y_l^j},$$

where $\delta_x$ is the Dirac measure located at $x$ and $\nu^j(y_l^j)$ is equal to the probability of sampling the point $y_l^j$ from measure $\nu^j$. We denote the total cardinality of the support of all measures $\nu^j$ by

$$m = \sum_{j=1}^{m} m_j.$$

Fix any $\psi \in L_1(\boldsymbol{\nu})$. Suppose we are given access to $n$ i.i.d. samples $X_1, \ldots, X_n$ from a probability measure $\mu'_{\boldsymbol{\psi}}$ that satisfies

$$\|\mu_{\boldsymbol{\psi}} - \mu'_{\boldsymbol{\psi}}\|_{\mathrm{TV}} \le \varepsilon_\mu.$$

Then, for $j = 1, \ldots, k$ and $l = 1, \ldots, m_j$ consider

$$\widehat{\nu}^j(y_i^j) = \nu^j(y_i^j)\frac{1}{n}\sum_{i=1}^{n}\exp\left(\frac{\phi_{\psi^j}(X_i) + \psi^j(y) - c(x,y)}{\lambda}\right)$$

and for any parameter $\zeta \in (0, 1/2]$ define

$$\widetilde{\nu}^j = (1 - \zeta)\widehat{\nu}^j + \zeta\nu^j. \tag{13}$$

We claim that $\widetilde{\nu}^j$ implements the approximate Sinkhorn oracle with accuracy parameter of order $O(\varepsilon_\mu^{1/4})$ provided that $n$ is large enough. This is shown in the following lemma, the proof of which can be found in Appendix D.

**Lemma 2.** *Fix any $\delta \in (0, 1)$ and consider the setup described above. With probability at least $1 - \delta$, for each $j \in \{1, \ldots, k\}$ it holds simultaneously that the measure $\widetilde{\nu}^j$ defined in (13) satisfies all the properties listed in Definition 1 with accuracy parameter*

$$\varepsilon_j \le c_\infty(\mathcal{X})\left(2\zeta + \frac{1}{\zeta}m_j\varepsilon_\mu + \frac{1}{\zeta}m_j\sqrt{\frac{2\log\left(\frac{2m}{\delta}\right)}{n}}\right)^{1/2}.$$

The above lemma shows that a step of Algorithm 2 can be implemented provided access to i.i.d. sampling from some measure $\mu'_{\boldsymbol{\psi}}$ close to $\mu_{\boldsymbol{\psi}}$ in total variation norm, where $\psi$ is an arbitrary iterate of Algorithm 2. The remainder of this section is dedicated to showing that this can be achieved by sampling via Langevin Monte Carlo.

Henceforth, fix $\pi_{\mathrm{ref}}$ to be the Lebesgue measure on $\mathcal{X}$, which corresponds to the free-support barycenters setup. Then, for any $\psi$ we have

$$\mu_{\boldsymbol{\psi}}(dx) \propto \mathbb{1}_{\mathcal{X}}\exp(-V_{\boldsymbol{\psi}}(x)/\tau)dx, \quad \text{where} \quad V_{\boldsymbol{\psi}}(x) = \sum_{j=1}^{k} w_j\phi_{\psi^j}^j,$$

where $\mathbb{1}_{\mathcal{X}}$ is equal to one on $\mathcal{X}$ and zero everywhere else. It follows by (6) that $\|V_{\boldsymbol{\psi}}\|_{\mathrm{osc}} \le c_\infty(\mathcal{X})/\tau$. Further, let $\mathrm{diam}\mathcal{X} = \sup_{x,x'\in\mathcal{X}}\|x - x'\|_2$. By the convexity of $\mathcal{X}$, the uniform measure on $\mathcal{X}$ satisfies the logarithmic Sobolev inequality (LSI) with constant $\mathrm{diam}(\mathcal{X})^2/4$ (cf. [37]). Hence, by the Holley-Stroock perturbation argument [32], the measure $\mu_{\boldsymbol{\psi}}$ satisfies LSI with constant at most $\exp(2c_\infty(\mathcal{X})/\tau)\mathrm{diam}(\mathcal{X})^2/4 < \infty$.

It is well-established that Langevin Monte Carlo algorithms offer convergence guarantees for approximate sampling from a target measure subject to functional inequality constraints provided additional conditions hold such as the smoothness of the function $V_{\boldsymbol{\psi}}$. However, such guarantees do not directly apply to the measure $\mu_{\boldsymbol{\psi}}$ due to its constrained support. Instead, it is possible to approximate $\mu_{\boldsymbol{\psi}}$ arbitrarily well in total variation norm by a family of measures $(\mu_{\boldsymbol{\psi},\sigma})_{\sigma>0}$ (see Appendix E for details) supported on all of $\mathbb{R}^d$. Tuning the parameter $\sigma$ allows us to trade-off between the approximation quality of $\mu_{\boldsymbol{\psi},\sigma}$ and its LSI constant. Crucially, standard sampling guarantees for Langevin Monte Carlo (e.g., [57]) apply to the regularized measures $\mu_{\boldsymbol{\psi},\sigma}$, which leads to provable

guarantees for an implementation of Algorithm 2, thus furnishing the first convergence guarantees for computation of Wasserstein barycenters in the free support setup; see Theorem 3 stated below.

The above approximation argument applies to any cost function $c$ that is Lipschitz on $\mathcal{X}$ and exhibits quadratic growth at infinity. For the sake of simplicity, we consider the quadratic cost $c(x, y) = \|x - y\|_2^2$. The exact problem setup where we are able to obtain computational guarantees for free-support barycenter computation via Langevin Sampling is formalized below.

**Problem Setting 1.** Consider the setting described at the beginning of Section 4.1. In addition, suppose that

1. the reference measure $\pi_{\mathrm{ref}}(dx) = \mathbb{1}_{\mathcal{X}} dx$ is the Lebesgue measure supported on $\mathcal{X}$ (free-support setup);

2. it holds that $\mathcal{X} \subseteq \mathcal{B}_R = \{x \in \mathbb{R}^d : \|x\|_2 \leq R\}$ for some constant $R < \infty$;

3. the cost function $c : \mathbb{R}^d \times \mathbb{R}^d \to [0, \infty)$ is defined by $c(x, y) = \|x - y\|_2^2$;

4. for any $\psi$ we have access to a stationary point $x_\psi$ of $V_\psi$ over $\mathcal{X}$.

The last condition can be implemented in polynomial time using a first order gradient method. For our purposes, this condition is needed to obtain a good initialization point for the Unadjusted Langevin Algorithm following the explanation in [57, Lemma 1]; see Appendix E for further details.

We now proceed to the main result of this section, the proof of which can be found in Appendix E. The following theorem provides the first provably convergent method for computing Wasserstein barycenters in the free-support setting. We remark that a stochastic approximation argument of a rather different flavor used to compute fixed-support Wasserstein barycenters (for $\tau \geq \lambda$) has been previously analyzed in [5].

**Theorem 3.** *Consider the setup described in Problem Setting 1. Then, for any confidence parameter $\delta \in (0, 1)$ and any accuracy parameter $\varepsilon > 0$, we can simulate a step of Algorithm 2 with success probability at least $1 - \delta$ in time polynomial in*

$$\varepsilon^{-1}, d, R, \exp(R^2/\tau), (Rd^{-1/4})^d, \tau^{-1}, \lambda^{-1}, d, m, \log(m/\delta).$$

*In particular, an $\varepsilon$-approximation of the $(\lambda, \tau)$-Barycenter can be obtained within the same computational complexity.*

Comparing the above guarantee with the discussion following the statement of Lemma 2, we see an additional polynomial dependence on $(Rd^{-1/4})^d$ (note that for $R \leq d^{1/4}$ this term disappears). We believe this term to be an artefact of our analysis appearing due to the approximation argument described above. Considering the setup with $R \leq d^{1/4}$, the running time of our algorithm depends exponentially in $R^2/\tau$.

We conclude with two observations. First, since approximating Wasserstein barycenters is generally NP-hard [4], an algorithm with polynomial dependence on all problem parameters does not exist if $P \neq NP$. Second, notice that computing an $\varepsilon$ approximation of $(\lambda, \tau)$-Barycenter can be done in time polynomial in $\varepsilon^{-1}$. This should be contrasted with numerical schemes based on discretizations of the set $\mathcal{X}$, which would, in general, result in computational complexity of order $(R/\varepsilon)^d$ to reach the same accuracy.

## 5 Conclusion

We introduced algorithms to compute doubly regularized entropic Wasserstein barycenters and studied their computational complexity, both in the fixed-support and in the free-support settings. Although a naive adaptation of the usual alternate maximization scheme from [7] to our setting leads to diverging iterates (at least for small values of $\tau$), our analysis shows that it is sufficient to damp these iterations to get a converging algorithm.

While we have focused on the problem of barycenters of measures, we note that the idea of entropic regularization is pervasive in other applications of optimal transport. There, the flexibility offered by the double entropic regularization may prove to be useful as well, and we believe that our damped algorithm could be adapted to these more general settings.

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

# A  Proof of Lemma 1

To simplify the notation throughout this proof, for each $j \in \{1, \ldots, k\}$ denote $\phi^j = \phi_{\psi^j}$. We have

$$E_{\lambda,\tau}^{\boldsymbol{\nu},w}(\boldsymbol{\psi}^*) - E_{\lambda,\tau}^{\boldsymbol{\nu},w}(\boldsymbol{\psi}) = \sum_{j=1}^{k} w_j \mathbf{E}_{Y \sim \nu^j} \left[ (\psi^*)^j(Y) - \psi^j(Y) \right] - \tau \log \frac{Z_{\boldsymbol{\psi}^*}}{Z_{\boldsymbol{\psi}}}. \tag{14}$$

Observe that for any $x \in \mathcal{X}$ it holds that

$$\frac{d\mu_{\boldsymbol{\psi}}}{d\mu_{\boldsymbol{\psi}^*}}(x) = \frac{Z_{\boldsymbol{\psi}^*}}{Z_{\boldsymbol{\psi}}} \exp\left( -\frac{\sum_{j=1}^{k} w_j(\phi^j(x) - (\phi^*)^j(x))}{\tau} \right).$$

Hence,

$$\begin{aligned}
\tau \log \frac{Z_{\boldsymbol{\psi}^*}}{Z_{\boldsymbol{\psi}}} &= \tau \log \mathbf{E}_{X \sim \mu_{\boldsymbol{\psi}^*}} \left[ \frac{Z_{\boldsymbol{\psi}^*}}{Z_{\boldsymbol{\psi}}} \right] \\
&= \tau \log \mathbf{E}_{X \sim \mu_{\boldsymbol{\psi}^*}} \left[ \frac{d\mu_{\boldsymbol{\psi}}}{d\mu_{\boldsymbol{\psi}^*}}(x) \exp\left( \frac{\sum_{j=1}^{k} w_j(\phi^j(x) - (\phi^*)^j(x))}{\tau} \right) \right] \\
&= \tau \log \mathbf{E}_{X \sim \mu_{\boldsymbol{\psi}}} \left[ \exp\left( \frac{\sum_{j=1}^{k} w_j(\phi^j(x) - (\phi^*)^j(x))}{\tau} \right) \right] \\
&= \sup_{\mu \ll \mu_{\boldsymbol{\psi}}} \left\{ \mathbf{E}_{X \sim \mu} \left[ \sum_{j=1}^{k} w_j(\phi^j(x) - (\phi^*)^j(x)) \right] - \tau \mathrm{KL}(\mu, \mu_{\boldsymbol{\psi}}) \right\}, \tag{15}
\end{aligned}$$

where in the final expression we have applied the Donsker-Varadhan variational principle (i.e., convex-conjugate duality between KL-divergence and cumulant generating functions); therein, the supremum runs over probability measures $\mu$ absolutely continuous with respect to $\mu_{\boldsymbol{\psi}}$, and it is attained by $\mu$ defined as

$$\begin{aligned}
\mu(dx) &\propto \exp\left( \frac{1}{\tau} \sum_{j=1}^{k} w_j(\phi^j(x) - (\phi^*)^j(x)) \right) \mu_{\boldsymbol{\psi}}(dx) \\
&\propto \exp\left( \frac{1}{\tau} \sum_{j=1}^{k} w_j(\phi^j(x) - (\phi^*)^j(x)) \right) \exp\left( -\frac{1}{\tau} \sum_{j=1}^{k} w_j \phi^j(x) \right) \pi_{\mathrm{ref}}(dx) \\
&\propto \exp\left( -\frac{1}{\tau} \sum_{j=1}^{k} w_j(\phi^*)^j(x) \right) \pi_{\mathrm{ref}}(dx) = \pi_{\boldsymbol{\psi}^*}(dx).
\end{aligned}$$

That is, the supremum in (15) is attained by $\mu = \mu_{\boldsymbol{\psi}^*}$. Hence, the identity (14) becomes

$$\begin{aligned}
E_{\lambda,\tau}^{\boldsymbol{\nu},w}&(\boldsymbol{\psi}^*) - E_{\lambda,\tau}^{\boldsymbol{\nu},w}(\boldsymbol{\psi}) \\
&= \sum_{j=1}^{k} w_j \mathbf{E}_{Y \sim \nu^j} \left[ (\psi^*)^j(Y) - \psi^j(Y) \right] - \mathbf{E}_{X \sim \mu_{\boldsymbol{\psi}^*}} \left[ \sum_{j=1}^{k} w_j(\phi^j(X) - (\phi^*)^j(X)) \right] \\
&\quad + \tau \mathrm{KL}(\mu_{\boldsymbol{\psi}^*}, \mu_{\boldsymbol{\psi}}) \\
&= \sum_{j=1}^{k} w_j \left( \mathbf{E}_{Y \sim \nu^j} \left[ (\psi^*)^j(Y) - \psi^j(Y) \right] + \mathbf{E}_{X \sim \mu_{\boldsymbol{\psi}^*}} \left[ (\phi^*)^j(X)) - \phi^j(X) \right] \right) + \tau \mathrm{KL}(\mu_{\boldsymbol{\psi}^*}, \mu_{\boldsymbol{\psi}}) \\
&\geq \tau \mathrm{KL}(\mu_{\boldsymbol{\psi}^*}, \mu_{\boldsymbol{\psi}}),
\end{aligned}$$

where the final inequality follows by noting that for each $j$ the optimality of the pair $((\phi^*)^j, (\psi^*)^j)$ for the entropic optimal transport dual objective $E_\lambda^{\mu_{\boldsymbol{\psi}^*}, \nu^j}$ implies that

$$\begin{aligned}
\mathbf{E}_{Y \sim \nu^j} &\left[ (\psi^*)^j(Y) - \psi^j(Y) \right] + \mathbf{E}_{X \sim \mu_{\boldsymbol{\psi}^*}} \left[ (\phi^*)^j(X) - \phi^j(X) \right] \\
&= E_\lambda^{\mu, \nu^j}((\phi^*)^j, (\psi^*)^j) - E_\lambda^{\mu, \nu^j}(\phi^j, \psi^j) \geq 0.
\end{aligned}$$

The proof of Lemma 1 is complete. $\qquad\square$

# B    Proof of Proposition 1

Recall that for any non-negative integer $t$ we have

$$\mu_t(dx) = Z_t^{-1} \exp\left(-\frac{\sum_{j=1}^{k} w_j \phi_t^j(x)}{\tau}\right) \pi_{\mathrm{ref}}(dx).$$

where $Z_t$ is the normalizing constant defined by

$$Z_t = \int_{\mathcal{X}} \exp\left(-\frac{\sum_{j=1}^{k} w_j \phi_t^j(x)}{\tau}\right) \pi_{\mathrm{ref}}(dx).$$

With the notation introduced above, we have

$$E(\boldsymbol{\psi}_t) = \sum_{j=1}^{k} w_j \mathbf{E}_{Y \sim \nu^j}\left[\psi_t^j(Y)\right] - \tau \log Z_t.$$

Hence,

$$E(\boldsymbol{\psi}_{t+1}) - E(\boldsymbol{\psi}_t) = \sum_{j=1}^{k} w_j \mathbf{E}_{Y \sim \nu^j}\left[\psi_{t+1}^j(Y) - \psi_t^j(Y)\right] - \tau \log \frac{Z_{t+1}}{Z_t}.$$

$$= \eta\lambda \sum_{j=1}^{k} w_j \mathbf{E}_{Y \sim \nu^j}\left[\log \frac{d\nu^j}{d\nu_t^j}(Y)\right] - \tau \log \frac{Z_{t+1}}{Z_t}.$$

$$= \min(\lambda, \tau) \sum_{j=1}^{k} w_j \mathrm{KL}(\nu^j, \nu_t^j) - \tau \log \frac{Z_{t+1}}{Z_t}.$$

Therefore, to prove Proposition 1 it suffices to show that the inequality

$$\log \frac{Z_{t+1}}{Z_t} \leq 0 \tag{16}$$

holds for any $t \geq 0$. We will complete the proof of Proposition 1 using the following lemma, the proof of which is deferred to the end of this section.

**Lemma 3.** *Let $(\boldsymbol{\psi}_t)_{t \geq 0}$ be any sequence of the form*

$$\psi_{t+1}^j = \psi_t^j + \eta\lambda \log(\Delta_t^j),$$

*where for $j \in \{1, \ldots, k\}$, $(\Delta_t^j)_{t \geq 0}$ is an arbitrary sequence of strictly positive functions and $\eta = \min(1, \tau/\lambda)$. Then, for any $t \geq 0$ it holds that*

$$\tau \log \frac{Z_{\boldsymbol{\psi}_{t+1}}}{Z_{\boldsymbol{\psi}_t}} \leq \min(\lambda, \tau) \log \sum_{j=1}^{k} w_j \mathbf{E}_{Y \sim \nu_{\boldsymbol{\psi}_t}^j}\left[\Delta_t^j(Y)\right].$$

To complete the proof of Proposition 1, we will apply the above lemma with $\Delta_t^j = \frac{d\nu^j}{d\nu_t^j}$. Indeed, we have

$$\tau \log \frac{Z_{t+1}}{Z_t} \leq \min(\lambda, \tau) \log \sum_{j=1}^{k} w_j \mathbf{E}_{Y \sim \nu_t^j}\left[\frac{d\nu^j}{d\nu_t^j}(Y)\right]$$

$$= \min(\lambda, \tau) \log \sum_{j=1}^{k} w_j \mathbf{E}_{Y \sim \nu^j}[1]$$

$$= 0.$$

By (16), the proof of Proposition 1 is complete. $\qquad\square$

## B.1 Proof of Lemma 3

We will break down the proof with the help of the following lemma, the proof of which can be found in Section B.2.

**Lemma 4.** *For any sequence $(\psi_t)_{t\geq 0}$ and any $t \geq 0$ it holds that*

$$
\log \frac{Z_{\psi_{t+1}}}{Z_{\psi t}} \leq \begin{cases} \frac{\lambda}{\tau} \log \sum_{j=1}^{k} w_j \mathbf{E}_{X\sim\mu_t} \left[ \exp \left( \frac{-\phi_{t+1}^j(X)+\phi_t^j(X)}{\tau} \right)^{\tau/\lambda} \right] & \text{if } \tau \geq \lambda, \\ \log \sum_{j=1}^{k} w_j \mathbf{E}_{X\sim\mu_t} \left[ \exp \left( \frac{-\phi_{t+1}^j(X)+\phi_t^j(X)}{\tau} \right) \right] & \text{if } \tau < \lambda, \end{cases}
$$

*where $\phi_t = \phi_{\psi_t}$ and $\mu_t(dx) = Z_{\psi_t}^{-1} \exp(-\sum_{j=1}^{k} w_j \phi_t^j(x)/\tau)\pi_{\text{ref}}(dx)$.*

Observe that the sequence $(\psi_t)_{t\geq 0}$ of the form stated in Lemma 3 satisfies, for any for any $j \in \{1,\ldots,k\}$ and any $t \geq 0$,

$$
\exp \left( \frac{-\phi_{t+1}^j + \phi_t^j}{\tau} \right) = \exp \left( -\frac{\lambda}{\tau} \log \frac{d\mu_t}{d\tilde{\mu}_t^j} \right) = \left( \frac{d\tilde{\mu}_t^j}{d\mu_t} \right)^{\lambda/\tau},
$$

where

$$
\frac{d\tilde{\mu}_t^j}{d\mu_t}(x) = \int_{\mathcal{X}} \nu(dy) \exp \left( \frac{\psi_{t+1}^j(y) + \phi_t^j(x) - c(x,y)}{\lambda} \right)
$$

$$
= \int_{\mathcal{X}} \nu^j(dy) \Delta_t^j(y)^{\eta} \exp \left( \frac{\psi_t^j(y) + \phi_t^j(x) - c(x,y)}{\lambda} \right).
$$

Hence, by Lemma 4 we have

$$
\log \frac{Z_{\psi_{t+1}}}{Z_{\psi_t}}
$$

$$
\leq \frac{1}{\tau} \min(\lambda,\tau) \sum_{j=1}^{k} w_j \mathbf{E}_{X\sim\mu_t} \Bigg[
$$

$$
\left( \int_{\mathcal{X}} \nu^j(dy) \Delta_t^j(y)^{\eta} \exp \left( \frac{\psi_t^j(y) + \phi_t^j(X) - c(X,y)}{\lambda} \right) \right)^{\max(1,\lambda/\tau)} \Bigg]. \tag{17}
$$

We split the remaining proof into two cases: $\tau \geq \lambda$ and $\tau < \lambda$.

**The case $\tau \geq \lambda$.** When $\tau \geq \lambda$, we have $\max(1, \lambda/\tau) = 1$ and $\eta = \min(1, \tau/\lambda) = 1$. Thus, (17) yields

$$
\log \frac{Z_{\psi_{t+1}}}{Z_{\psi_t}}
$$

$$
\leq \frac{1}{\tau} \min(\lambda,\tau) \log \sum_{j=1}^{k} w_j \mathbf{E}_{X\sim\mu_t} \left[ \int_{\mathcal{X}} \nu^j(dy) \Delta_t^j(y) \exp \left( \frac{\psi_t^j(y) + \phi_t^j(X) - c(X,y)}{\lambda} \right) \right]
$$

$$
= \frac{1}{\tau} \min(\lambda,\tau) \log \sum_{j=1}^{k} w_j \left[ \int_{\mathcal{X}} \mu_t(dx) \int_{\mathcal{X}} \nu^j(dy) \Delta_t^j(y) \exp \left( \frac{\psi_t^j(y) + \phi_t^j(X) - c(X,y)}{\lambda} \right) \right]
$$

$$
= \frac{1}{\tau} \min(\lambda,\tau) \log \sum_{j=1}^{k} w_j \left[ \int_{\mathcal{X}} \Delta_t^j(y)\nu^j(dy) \int_{\mathcal{X}} \exp \left( \frac{\psi_t^j(y) + \phi_t^j(X) - c(X,y)}{\lambda} \right) \mu_t(dx) \right]
$$

$$
= \frac{1}{\tau} \min(\lambda,\tau) \log \sum_{j=1}^{k} w_j \left[ \int_{\mathcal{X}} \Delta_t^j(y)\nu^j(dy) \frac{d\nu_{\psi_t}^j}{d\nu^j}(y) \right]
$$

$$
= \frac{1}{\tau} \min(\lambda,\tau) \log \sum_{j=1}^{k} w_j \mathbf{E}_{Y\sim\nu_{\psi_t}^j} \left[ \Delta_t^j(y) \right].
$$

This completes the proof of Lemma 3 when $\tau \geq \lambda$.

**The case $\tau < \lambda$.** For $j \in \{1, \dots, k\}$ and any $x \in \mathcal{X}$ define the measure $\rho_x$ by

$$\rho_x^j(dy) = \nu^j(dy) \exp\left(\frac{\psi_t^j(y) + \phi_t^j(x) - c(x,y)}{\lambda}\right).$$

By the definition of $\phi_t^j$, we have

$$\int_{\mathcal{X}} \rho_x^j(dy)$$

$$= \int_{\mathcal{X}} \nu(dy) \exp\left(\frac{\psi_t^j(y) - c(x,y)}{\lambda}\right) \exp\left(\frac{\phi_t^j(x)}{\lambda}\right)$$

$$= \int_{\mathcal{X}} \nu(dy) \exp\left(\frac{\psi_t^j(y) - c(x,y)}{\lambda}\right) \exp\left(-\log \int_{\mathcal{X}} \nu^j(dy') \exp\left(\frac{\psi_t^j(y') - c(x,y')}{\lambda}\right)\right)$$

$$= 1$$

In particular, $\rho_x$ is a probability measure. Hence, (17) can be rewritten as

$$\log \frac{Z_{\boldsymbol{\psi}_{t+1}}}{Z_{\boldsymbol{\psi}_t}} \leq \log \sum_{j=1}^k w_j \mathbf{E}_{X \sim \mu_t}\left[\mathbf{E}_{Y \sim \rho_X^j}\left[\Delta_t^j(Y)^\eta \Big| X\right]^{\lambda/\tau}\right]$$

Because $\lambda/\tau > 1$, the function $x \mapsto x^{\lambda/\tau}$ is convex. Applying Jensen's inequality to the conditional expectation and using the fact that $\eta\lambda/\tau = 1$, it follows that

$$\log \frac{Z_{\boldsymbol{\psi}_{t+1}}}{Z_{\boldsymbol{\psi}_t}} \leq \log \sum_{j=1}^k w_j \mathbf{E}_{X \sim \mu_t}\left[\mathbf{E}_{Y \sim \rho_X^j}\left[\Delta_t^j(Y)\Big| X\right]\right]$$

$$= \log \sum_{j=1}^k w_j \int_{\mathcal{X}} \mu_t(dx) \int_{\mathcal{X}} \Delta_t^j(y) \exp\left(\frac{\psi_t^j(y) + \phi_t^j(x) - c(x,y)}{\lambda}\right) \nu(dy). \quad (18)$$

By the definition of $\nu_{\boldsymbol{\psi}_t}^j$ we have

$$\frac{d\nu_{\boldsymbol{\psi}_t}^j}{d\nu^j}(y) = \int_{\mathcal{X}} \exp\left(\frac{\psi_t^j(y) + \phi_t^j(x) - c(x,y)}{\lambda}\right) \mu_t(dx).$$

Interchanging the order of integration in (18) and plugging in the above equation yields

$$\log \frac{Z_{\boldsymbol{\psi}_{t+1}}}{Z_{\boldsymbol{\psi}_t}} \leq \log \sum_{j=1}^k w_j \int_{\mathcal{X}} \left[\int_{\mathcal{X}} \exp\left(\frac{\psi_t^j(y) + \phi_t^j(x) - c(x,y)}{\lambda}\right) \mu_t(dx)\right] \Delta_t^j(y)\nu^j(dy)$$

$$= \log \sum_{j=1}^k w_j \int_{\mathcal{X}} \left[\frac{d\nu_{\boldsymbol{\psi}_t}^j}{d\nu^j}(y)\right] \Delta_t^j(y)\nu^j(dy)$$

$$= \log \sum_{j=1}^k w_j \mathbf{E}_{Y \sim \nu_{\boldsymbol{\psi}_t}^j}\left[\Delta_t^j(Y)\right].$$

This completes the proof of Lemma 3. $\qquad\square$

### B.2 Proof of Lemma 4

To simplify the notation, denote $Z_t = Z_{\boldsymbol{\psi}_t}$. Let $x \in \mathcal{X}$ and $t \geq 0$. We have $\mu_t \ll \mu_{t+1}$ with the Radon-Nikodym derivative $d\mu_{t+1}/d\mu_t$ given by

$$\frac{d\mu_{t+1}}{d\mu_t}(x) = \frac{Z_t}{Z_{t+1}} \exp\left(\frac{-\sum_{j=1}^k w_k(\phi_{t+1}^j(x) - \phi_t^j(x))}{\tau}\right)$$

$$= \frac{Z_t}{Z_{t+1}} \prod_{j=1}^k \exp\left(\frac{-\phi_{t+1}^j(x) + \phi_t^j(x)}{\tau}\right)^{w_j}.$$

Multiplying both sides by $Z_{t+1}/Z_t$ and taking expectations with respect to $\mu_t$ yields

$$
\begin{aligned}
\frac{Z_{t+1}}{Z_t} &= \mathbf{E}_{X \sim \mu_{t+1}} \left[ \frac{Z_{t+1}}{Z_t} \right] \\
&= \mathbf{E}_{X \sim \mu_t} \left[ \frac{Z_{t+1}}{Z_t} \frac{d\mu_{t+1}}{d\mu_t}(X) \right] \\
&= \mathbf{E}_{X \sim \mu_t} \left[ \prod_{j=1}^{k} \exp\left( \frac{-\phi_{t+1}^j(X) + \phi_t^j(X)}{\tau} \right)^{w_j} \right].
\end{aligned}
$$

In the case $\tau < \lambda$, the proof is complete by the Arithmetic-Geometric mean inequality (recall that $w_j > 0$ for $j = 1, \ldots, k$ and $\sum_{j=1}^{k} w_j = 1$). On the other hand, if $\tau \geq \lambda$ then $x \mapsto x^{\lambda/\tau}$ is concave. Hence, it follows that

$$
\begin{aligned}
\log \frac{Z_{t+1}}{Z_t} &= \log \mathbf{E}_{X \sim \mu_t} \left[ \left( \prod_{j=1}^{k} \exp\left( \frac{-\phi_{t+1}^j(X) + \phi_t^j(X)}{\tau} \right)^{w_j \tau/\lambda} \right)^{\lambda/\tau} \right] \\
&\leq \log \mathbf{E}_{X \sim \mu_t} \left[ \left( \prod_{j=1}^{k} \exp\left( \frac{-\phi_{t+1}^j(X) + \phi_t^j(X)}{\tau} \right)^{w_j \tau/\lambda} \right) \right]^{\lambda/\tau} \\
&= \frac{\lambda}{\tau} \log \mathbf{E}_{X \sim \mu_t} \left[ \left( \prod_{j=1}^{k} \exp\left( \frac{-\phi_{t+1}^j(X) + \phi_t^j(X)}{\tau} \right)^{w_j \tau/\lambda} \right) \right] \\
&\leq \sum_{j=1}^{k} w_j \mathbf{E}_{X \sim \mu_t} \left[ \exp\left( \frac{-\phi_{t+1}^j(X) + \phi_t^j(X)}{\tau} \right)^{\tau/\lambda} \right],
\end{aligned}
$$

where the final step follows via the Arithmetic-Geometric mean inequality. This completes the proof of Lemma 4. □

## C   Proof of Theorem 2

For every $t \geq 0$ and $j \in \{1, \ldots, k\}$, let $\widetilde{\nu}_t^j$ be the distribution returned by the approximate Sinkhorn oracle that satisfies the properties listed in Definition 1. We follow along the lines of proof of Theorem 1.

First, we will establish an upper bound on the oscillation norm of the iterates $\widetilde{\psi}_t$. Indeed, by the property four in Definition 1 we have

$$
\|\widetilde{\psi}_{t+1}^j\|_{\mathrm{osc}} \leq (1-\eta)\|\widetilde{\psi}_t^j\|_{\mathrm{osc}} + \eta c_\infty(\mathcal{X}).
$$

Since $\widetilde{\psi}_0^j = 0$, for any $t \geq 0$ we have $\|\widetilde{\psi}_t^j\|_{\mathrm{osc}} \leq c_\infty(\mathcal{X})$.

Let $\widetilde{\delta}_t = E_{\lambda,\tau}^{\boldsymbol{\nu},w}(\psi^*) - E_{\lambda,\tau}^{\boldsymbol{\nu},w}(\widetilde{\psi}_t)$ be the suboptimality gap at time $t$. Using the concavity upper bound (10) and the property two in Definition 1 we have

$$
\begin{aligned}
\widetilde{\delta}_t &\leq 2c_\infty(\mathcal{X}) \sum_{j=1}^{k} w_j \|\nu^j - \nu_t^j\|_{\mathrm{TV}} \\
&\leq \varepsilon + 2c_\infty(\mathcal{X}) \sum_{j=1}^{k} w_j \|\nu^j - \widetilde{\nu}_t^j\|_{\mathrm{TV}} \\
&\leq \varepsilon + \sqrt{2}c_\infty(\mathcal{X}) \sum_{j=1}^{k} w_j \sqrt{\mathrm{KL}(\nu^j, \widetilde{\nu}_t^j)} \\
&\leq \varepsilon + \sqrt{2}c_\infty(\mathcal{X}) \sqrt{\sum_{j=1}^{k} w_j \mathrm{KL}(\nu^j, \widetilde{\nu}_t^j)}.
\end{aligned}
$$

Combining the property three stated in the Definition 1 with Lemma 3 we obtain

$$\tilde{\delta}_t - \tilde{\delta}_{t+1} \geq \min(\lambda, \tau) \sum_{j=1}^{k} w_j \mathrm{KL}(v^j, \tilde{v}_t^j) - \min(\lambda, \tau) \log\left(\sum_{j=1}^{k} w_j \int_{\mathcal{X}} \frac{d\nu_t}{d\tilde{\nu}_t}(y) \nu^j(dy)\right)$$

$$\geq \sum_{j=1}^{k} w_j \mathrm{KL}(v^j, \tilde{v}_t^j) - \min(\lambda, \tau) \log\left(1 + \varepsilon^2/(2c_\infty(\mathcal{X})^2)\right)$$

$$\geq \min(\lambda, \tau) \sum_{j=1}^{k} w_j \mathrm{KL}(v^j, \tilde{v}_t^j) - \frac{\min(\lambda, \tau)}{2c_\infty(\mathcal{X})^2} \varepsilon^2$$

$$\geq \frac{\min(\lambda, \tau)}{2c_\infty(\mathcal{X})^2} \max\left\{0, \tilde{\delta}_t - \varepsilon\right\}^2 - \frac{\min(\lambda, \tau)}{2c_\infty(\mathcal{X})^2} \varepsilon^2.$$

Provided that $\tilde{\delta}_t \geq 2\varepsilon$ it holds that

$$(\tilde{\delta}_t - 2\varepsilon) - (\tilde{\delta}_{t+1} - 2\varepsilon) \geq \frac{\min(\lambda, \tau)}{2c_\infty(\mathcal{X})}(\tilde{\delta}_t - 2\varepsilon)^2.$$

Let $T$ be the first index such that $\tilde{\delta}_{T+1} < 2\varepsilon$ and set $T = \infty$ if no such index exists. Then, the above equation is valid for any $t \leq T$. In particular, repeating the proof of Theorem 1, for any $t \leq T$ we have

$$\tilde{\delta}_t - 2\varepsilon \leq \frac{2c_\infty(\mathcal{X})^2}{\min(\lambda, \tau)} \frac{1}{t},$$

which completes the proof of this theorem. $\qquad\square$

# D  Proof of Lemma 2

The first property – the positivity of the probability mass function of $\tilde{\nu}^j$ – is immediate from its definition.

To simplify the notation, denote in what follows

$$K^j(x, y) = \exp\left(\frac{\phi_{\psi^j}(x) + \psi^j(y) - c(x, y)}{\lambda}\right).$$

With this notation, recall that

$$\widehat{\nu}_{\boldsymbol{\psi}}^j(y_l^j) = \frac{1}{n} \sum_{i=1}^{n} \nu^j(y_l^j) K(X_i, y_l^j).$$

The above is a sum of $n$ non-negative random variables bounded by one with expectation

$$(\nu')^j(y_l^j) = \mathbf{E}_{X \sim \mu_{\boldsymbol{\psi}}'}\left[\nu^j(y_l^j)\right]$$

It follows by Hoeffding's inequality and the union bound that with probability at least $1 - \delta$ the following holds for any $j \in \{1, \ldots, k\}$ and any $l \in \{1, \ldots, m_j\}$:

$$\left|\widehat{\nu}_{\boldsymbol{\psi}}(y_l^j) - (\nu')^j(y_l^j)\right| \leq \sqrt{\frac{2\log\left(\frac{2m}{\delta}\right)}{n}}.$$

In particular, the above implies that

$$\|\tilde{\nu}_{\boldsymbol{\psi}}^j - \nu_{\boldsymbol{\psi}}^j\|_{\mathrm{TV}} \leq 2\zeta + (1 - \zeta)\|\tilde{\nu}_{\boldsymbol{\psi}}^j - \nu_{\boldsymbol{\psi}}^j\|_{\mathrm{TV}}$$

$$\leq 2\zeta + (1 - \zeta)\|\tilde{\nu}_{\boldsymbol{\psi}}^j - (\nu')^j\|_{\mathrm{TV}} + (1 - \zeta)\|(\nu')^j - \nu_{\boldsymbol{\psi}}^j\|_{\mathrm{TV}}$$

$$\leq 2\zeta + \|\tilde{\nu}_{\boldsymbol{\psi}}^j - (\nu')^j\|_{\mathrm{TV}} + \|(\nu')^j - \nu_{\boldsymbol{\psi}}^j\|_{\mathrm{TV}}$$

$$\leq 2\zeta + m_j \varepsilon_\mu + m_j \sqrt{\frac{2\log\left(\frac{2m}{\delta}\right)}{n}}.$$

Notice that the above bound can be made arbitrarily close to $m_j \varepsilon_\mu$ by taking a large enough $n$ and a small enough $\zeta$. This proves the second property of Definition 1.

To prove the third property, observe that

$$\mathbf{E}_{Y \sim \nu^j} \left[ \frac{\nu_{\boldsymbol{\psi}}^j(Y)}{\widetilde{\nu}_{\boldsymbol{\psi}}^j(Y)} \right] = \mathbf{E}_{Y \sim \nu^j} \left[ \frac{\widehat{\nu}_{\boldsymbol{\psi}}^j(Y)}{\widetilde{\nu}_{\boldsymbol{\psi}}^j(Y)} + \frac{\nu_{\boldsymbol{\psi}}^j(Y) - \widehat{\nu}_{\boldsymbol{\psi}}^j(Y)}{\widetilde{\nu}_{\boldsymbol{\psi}}^j(Y)} \right]$$

$$\leq \mathbf{E}_{Y \sim \nu^j} \left[ \frac{1}{1 - \zeta} + \frac{\nu_{\boldsymbol{\psi}}^j(Y) - \widehat{\nu}_{\boldsymbol{\psi}}^j(Y)}{\widetilde{\nu}_{\boldsymbol{\psi}}^j(Y)} \right]$$

$$\leq \mathbf{E}_{Y \sim \nu^j} \left[ 1 + \frac{\zeta}{1 - \zeta} + \frac{\left| \nu_{\boldsymbol{\psi}}^j(Y) - \widehat{\nu}_{\boldsymbol{\psi}}^j(Y) \right|}{\widetilde{\nu}_{\boldsymbol{\psi}}^j(Y)} \right]$$

$$\leq 1 + \frac{\zeta}{1 - \zeta} + \frac{1}{\zeta} \| \nu_{\boldsymbol{\psi}}^j(Y) - \widehat{\nu}_{\boldsymbol{\psi}}^j(Y) \|_{\mathrm{TV}}$$

$$\leq 1 + 2\zeta + \frac{1}{\zeta} \left( m_j \varepsilon_\mu + m_j \sqrt{\frac{2 \log\left(\frac{2m}{\delta}\right)}{n}} \right).$$

This concludes the proof of the third property.

It remains to prove the fourth property of Definition 1. Observe that for any $y, y'$ we have

$$\left( \psi^j(y) - \eta \lambda \log \frac{\widetilde{\nu}^j(y)}{\nu^j(y)} \right) - \left( \psi^j(y') - \eta \lambda \log \frac{\widetilde{\nu}^j(y')}{\nu^j(y')} \right)$$

$$= \left( \psi^j(y) - \psi^j(y') \right) + \eta \lambda \log \left( \frac{\zeta + (1 - \zeta)\frac{1}{n} \sum_{i=1}^n K^j(X_i, y')}{\zeta + (1 - \zeta)\frac{1}{n} \sum_{i=1}^n K^j(X_i, y)} \right)$$

$$= \left( \psi^j(y) - \psi^j(y') \right) + \eta \lambda \log \left( \frac{\frac{\zeta}{1-\zeta} + \frac{1}{n} \sum_{i=1}^n K^j(X_i, y')}{\frac{\zeta}{1-\zeta} + \frac{1}{n} \sum_{i=1}^n K^j(X_i, y)} \right)$$

$$\leq \left( \psi^j(y) - \psi^j(y') \right) + \eta \lambda \log \left( \frac{\frac{\zeta}{1-\zeta} + \exp\left( \frac{c_\infty(\mathcal{X}) + \psi^j(y') - \psi^j(y)}{\lambda} \right) \frac{1}{n} \sum_{i=1}^n K^j(X_i, y)}{\frac{\zeta}{1-\zeta} + \frac{1}{n} \sum_{i=1}^n K^j(X_i, y)} \right).$$

Now observe that for any $a, b > 0$ the function $g : [0, \infty) \to (0, \infty)$ defined by $g(x) = (x+a)/(x+b)$ is increasing if $a < b$ and decreasing if $a \geq b$. Thus, $g$ is maximized either at zero or at infinity. It thus follows that

$$\eta \lambda \log \left( \frac{\frac{\zeta}{1-\zeta} + \exp\left( \frac{c_\infty(\mathcal{X})}{\lambda} \right) \frac{1}{n} \sum_{i=1}^n K^j(X_i, y)}{\frac{\zeta}{1-\zeta} + \frac{1}{n} \sum_{i=1}^n K^j(X_i, y)} \right)$$

$$\leq \begin{cases} \eta c_\infty(\mathcal{X}) - \eta(\psi^j(y) - \psi^j(y')) & \text{if } \exp\left( \frac{c_\infty(\mathcal{X})}{\lambda} \right) \geq 1 \\ 0 & \text{otherwise.} \end{cases}$$

This proves the claim and completes the proof of this lemma. $\qquad \square$

# E   Proof of Theorem 3

The purpose of this section is to show how sampling via Langevin Monte Carlo algorithm yields the first provable convergence guarantees for computing barycenters in the free-support setup (cf. the discussion at the end of Section 2.2). In particular, we provide computational guarantees for implementing Algorithm 2.

A measure $\mu$ is said to satisfy the logarithmic Sobolev inequality (LSI) with constant $C$ if for all sufficiently smooth functions $f$ it holds that

$$\mathbf{E}_\mu[f^2 \log f^2] - \mathbf{E}_\mu[f^2] \log \mathbf{E}_\mu[g^2] \leq 2C \mathbf{E}_\mu[\|\nabla f\|_2^2].$$

To sample from a measure $\mu(dx) = \exp(-f(x))dx$ supported on $\mathbb{R}^d$, the unadjusted Langevin Monte Carlo algorithm is defined via the following recursive update rule:

$$x_{k+1} = x_k - \eta\nabla f(x_k) + \sqrt{2\eta}Z_k, \quad \text{where} \quad Z_k \sim \mathcal{N}(0, I_d). \tag{19}$$

The following Theorem is due to Vempala and Wibisono [57, Theorem 3].

**Theorem 4.** *Let $\mu(dx) = \exp(-f(x))dx$ be a measure on $\mathbb{R}^d$. Suppose that $\mu$ satisfies LSI a with constant $C$ and that $f$ has L-Lipschitz gradient with respect to the Euclidean norm. Consider the sequence of iterates $(x_k)_{k\geq 0}$ defined via (19) and let let $\rho_k$ be the distribution of $x_k$. Then, for any $\varepsilon > 0$, any $\eta \leq \frac{1}{8L^2C}\min\{1, \frac{\varepsilon}{4d}\}$, and any $k \geq \frac{2C}{\eta}\log\frac{2\mathrm{KL}(\rho_0,\mu)}{\varepsilon}$, it holds that*

$$\mathrm{KL}(\rho_k, \mu) \leq \varepsilon.$$

Thus, LSI on the measure $\mu$ provides convergence guarantees on $\mathrm{KL}(\rho_k, \mu)$. It is shown in [57, Lemma 1] how to initialize the iterate $x_0$ so that $\mathrm{KL}(\rho_0, \mu)$ scales linearly with the ambient dimension $d$ up to some additional terms. The final condition described in Problem Setting 1 ensures that (by [57, Lemma 1]) for any $\sigma > 0$, the initialization scheme $x_0 \sim \mathcal{N}(x_\psi, I_d)$ for the Langevin algorithm (19) satisfies

$$\mathrm{KL}(\rho_0, \mu_{\psi,\sigma}) \leq \frac{c_\infty(\mathcal{X})}{\tau} + \frac{d}{2}\log\frac{L_\sigma}{2\pi},$$

where $L_\sigma$ is the smoothness constant of $V_\psi/\tau + \mathrm{dist}(x, \mathcal{X})/(2\sigma^2)$ (see Lemma 5) and $\mu_{\psi,\sigma}$ is the probability measure defined in (20).

To implement the approximate Sinkhorn oracle described in Definition 1, we can combine Lemma 2 with approximate sampling via Langevin Monte Carlo; note that by Pinsker's inequality, Kullback-Leibler divergence guarantees provide total variation guarantees which are sufficient for the application of Lemma 2. Therefore, providing provable convergence guarantees for Algorithm 2 amounts to proving that we can do arbitrarily accurate approximate sampling from distributions of the form

$$\mu_\psi(dx) \propto \mathbb{1}_\mathcal{X}(x)\exp(-V_\psi(x)/\tau)dx, \quad \text{where} \quad V_\psi(x) = \sum_{j=1}^{k} w_j\phi_{\psi^j}^j(x).$$

Here $\mathbb{1}_\mathcal{X}$ is the indicator function of $\mathcal{X}$, $\psi$ is an arbitrary iterate generated by Algorithm 2, and we consider the free-support setup characterized via the choice $\pi_{\mathrm{ref}}(dx) = \mathbb{1}_\mathcal{X}dx$.

Notice that we cannot apply Theorem 4 directly because the measure $\mu_\psi$ defined above has constrained support while Theorem 4 only applies for measures supported on all of $\mathbb{R}^d$. Nevertheless, we will show that the compactly supported measure $\mu_\psi$ can be approximated by a measure $\mu_{\psi,\sigma}$, where the parameter $\sigma$ will trade-off LSI constant of $\mu_{\psi,\sigma}$ against the total variation norm between the two measures. To this end, define

$$\mu_{\psi,\sigma} =\propto \exp(-V_\psi(x)/\tau - \mathrm{dist}(x, \mathcal{X})^2/(2\sigma^2))dx, \quad \text{where} \quad \mathrm{dist}(x, \mathcal{X}) = \inf_{y\in\mathcal{X}}\|x - y\|_2. \tag{20}$$

The following lemma, proved at the end of this section, collects the main properties of the measure $\mu_{\psi,\sigma}$.

**Lemma 5.** *Consider the setup described in Problem Setting 1. Let $\psi$ be any iterate generated by Algorithm 2 and let $\mu_{\psi,\sigma}$ be the distribution defined in (20). Then, the measure $\mu_{\psi,\sigma}$ satisfies the following properties:*

1. *For any $\sigma \in (0, 1/4]$ it holds that*

$$\|\mu_\psi - \mu_{\psi,\sigma}\|_{\mathrm{TV}} \leq 2\sigma\exp\left(\frac{8R^2}{\tau}\right)\left[\left(4Rd^{-1/4}\right)^{d-1} + 1\right].$$

2. *Let $V_\sigma(x) = \exp(-V_\psi(x)/\tau - \mathrm{dist}(x, \mathcal{X})^2/(2\sigma^2))$; thus $\mu_{\psi,\sigma}(dx) = \exp(-V_\sigma(x))dx$. The function $V_\sigma$ has $L_\sigma$-Lipschitz gradient where*

$$L_\sigma = \frac{1}{\tau} + \frac{1}{\tau\lambda}4R^2\max_j m_j + \frac{1}{\sigma^2}.$$

3. *The measure $\mu_{\psi,\sigma}$ satisfies LSI with a constant $C_\sigma = \mathrm{poly}(R, \exp(R^2/\tau), L_\sigma)$.*

Above, the notation $C = \mathrm{poly}(x, y, z)$ denotes a constant that depends polynomially on $x, y$ and $z$. With the above lemma at hand, we are ready to prove Theorem 3.

*Proof of Theorem 3.* Let $\psi$ be an arbitrary iterate generated via Algorithm 2. We can simulate a step of approximate Sinkhorn oracle with accuracy $\varepsilon$ via Lemma 2 (with $\zeta = \varepsilon/4$) in time $\mathrm{poly}(n, m, d)$ provided access to $n = \mathrm{poly}(\varepsilon^{-1}, m, \log(m/\delta))$ samples from any distribution $\mu'_\psi$ such that

$$\|\mu'_\psi - \mu_\psi\|_{\mathrm{TV}} \leq \frac{\varepsilon^2}{16m}. \tag{21}$$

To find a choice of $\mu'_\psi$ satisfying the above bound, consider the distribution

$$\mu_{\psi,\sigma} \quad \text{with} \quad \sigma = \frac{\varepsilon^2}{32m} \cdot \left( 2\exp\left(\frac{8R^2}{\tau}\right) \left[ \left(4Rd^{-1/4}\right)^{d-1} + 1 \right] \right)^{-1}.$$

Let $C_\sigma$ and $L_\sigma$ be the LSI and smoothness constants of the distribution $\mu_{\psi,\sigma}$ provided in Lemma 5. By Theorem 4, it suffices to run the Langevin algorithm (19) for $\mathrm{poly}(\varepsilon^{-1}, m, d, C_\sigma, L_\sigma)$ number of iterations to obtain a sample from a distribution $\widetilde{\mu}_{\psi,\sigma}$ such that

$$\|\widetilde{\mu}_{\psi,\sigma} - \mu_{\psi,\sigma}\|_{\mathrm{TV}} \leq \frac{\varepsilon^2}{32m}.$$

In particular, by the triangle inequality for the total variation norm, the choice $\mu'_\psi = \widetilde{\mu}_{\psi,\sigma}$ satisfies (21). This finishes the proof. $\qquad\square$

### E.1   Proof of Lemma 5

To simplify the notation, denote $\mu = \mu_\psi, \mu_\sigma = \mu_{\psi,\sigma}, V(x) = V_\psi(x)/\tau$, and $V_\sigma(x) = V(x)/\tau + \mathrm{dist}(x, \mathcal{X})^2/(2\sigma^2)$.

**Total variation norm bound.**   With the above shorthand notation, we have

$$\mu(dx) = \mathbb{1}_{\mathcal{X}} Z^{-1} \exp(-V(x))dx, \quad \text{where} \quad Z = \int_{\mathcal{X}} \exp(-V(x))dx$$

and

$$\mu_\sigma(dx) = (Z + Z_\sigma)^{-1} \exp(-V_\sigma(x))dx, \quad \text{where} \quad Z_\sigma = \int_{\mathbb{R}^d \setminus \mathcal{X}} \exp(-V_\sigma(x))dx.$$

We have

$$\|\mu - \mu_\sigma\|_{\mathrm{TV}} = \int_{\mathbb{R}^d \setminus \mathcal{X}} (Z + Z_\sigma)^{-1} \exp(-V_\sigma(x))dx + \int_{\mathcal{X}} |(Z + Z_\sigma)^{-1} - Z^{-1}| \exp(-V(x))dx$$

$$= \frac{2Z_\sigma}{Z + Z_\sigma} \leq \frac{2Z_\sigma}{Z} \leq 2\exp\left(\frac{c_\infty(\mathcal{X})}{\tau}\right) Z_\sigma \leq 2\exp\left(\frac{4R^2}{\tau}\right) Z_\sigma.$$

We thus need to upper bound $Z_\sigma$. Let $\mathrm{Vol}(A)$ be the Lebesgue measure of the set $A$, let $\partial A$ denote the boundary of $A$, and let $A + B = \{a + b : a \in A, b \in B\}$ be the Minkowski sum of sets $A$ and $B$. Using the facts that for each $j \in \{1, \ldots, k\}$ we have $\sup_{y \in \mathcal{X}} \psi^j(y) \leq c_\infty(\mathcal{X}) \leq 4R^2$ and that

$\mathcal{X} \subseteq \mathcal{B}_R = \{x : \|x\|_2 \leq R\}$ we have

$$
\begin{aligned}
Z_\sigma &= \int_{\mathbb{R}^d \setminus \mathcal{X}} \exp(-V_\sigma(x)) dx \\
&\leq \exp\left(\frac{4R^2}{\tau}\right) \int_{\mathbb{R}^d \setminus \mathcal{X}} \exp\left(-\frac{\mathrm{dist}(x, \mathcal{X})}{2\sigma^2}\right) dx \\
&= \exp\left(\frac{4R^2}{\tau}\right) \int_0^\infty \mathrm{Vol}(\partial(\mathcal{X} + \mathcal{B}_x)) \exp\left(-\frac{x^2}{2\sigma^2}\right) dx \\
&\leq \exp\left(\frac{4R^2}{\tau}\right) \int_0^\infty \mathrm{Vol}(\partial\mathcal{B}_{R+x}) \exp\left(-\frac{x^2}{2\sigma^2}\right) dx \\
&= \exp\left(\frac{4R^2}{\tau}\right) \frac{\pi^{d/2}}{\Gamma(d/2)} \int_0^\infty (R+x)^{d-1} \exp\left(-\frac{x^2}{2\sigma^2}\right) dx.
\end{aligned}
$$

Bounding $(R+x)^{d-1} \leq 2^{d-1}R^{d-1} + 2^{d-1}x^{d-1}$ and computing the integrals results in

$$
\begin{aligned}
\|\mu - \mu_\sigma\|_{\mathrm{TV}} &\leq 2\exp\left(\frac{8R^2}{\tau}\right) \frac{\pi^{d/2}}{\Gamma(d/2)} 2^{d-1} \left[R^{d-1}\sigma\frac{\sqrt{\pi}}{2} + 2^{d/2-1}\Gamma(d/2)\sigma^d\right] \\
&\leq 2\sigma\exp\left(\frac{8R^2}{\tau}\right) \left[\frac{(2R)^{d-1}}{\Gamma(d/2)} + (4\sigma)^{d-1}\right].
\end{aligned}
$$

Using the assumption $\sigma \leq 1/4$ and using the bound $\Gamma(d) \geq (d/2)^{d/2}$ we can further simplify the above bound to

$$
\|\mu - \mu_\sigma\|_{\mathrm{TV}} \leq 2\sigma\exp\left(\frac{8R^2}{\tau}\right) \left[\left(4Rd^{-1/4}\right)^{d-1} + 1\right],
$$

which completes the proof of the total variation bound.

**Lipschitz constant of the gradient.** Recall that for any any $j \in \{1, \ldots, d\}$ we have

$$
\phi^j(x) - \frac{1}{2}\|x\|_2^2 = -\lambda \log\left(\sum_{l=1}^{n_j} \exp\left(\frac{\psi^j(y_l^j) - \frac{\|y_l^j\|_2^2}{2} + \langle x, y_l^j \rangle}{\lambda}\right) \nu^j(y_l^j)\right).
$$

Denote $\widetilde{\phi}^j(x) = \phi^j(x) - \frac{1}{2}\|x\|_2^2$. Fix any $x, x'$ and define $g(t) = \widetilde{\phi}^j(x + (x' - x)t)$. Then, for any $t \in [0, 1]$ we have

$$
g''(s) = -\frac{1}{\lambda}\mathrm{Var}_{L\sim\rho_t}\left[(Y^j(x' - x))_L\right] \geq -\frac{1}{\lambda}\|x - x'\|_2^2 m_j 4R^2, \tag{22}
$$

where

$$
\rho_t(l) \propto \nu(y_l^j) \exp\left(\frac{\psi^j(y_l^j) - \frac{\|y_l^j\|_2^2}{2} + \langle x + t(x' - x), y_l^j \rangle}{\lambda}\right)
$$

and $Y^j \in \mathbb{R}^{d \times m_j}$ is the matrix whose $l$-th column is equal to the vector $y_l^j$.

Because $\widetilde{\psi}^j$ is concave, the bound (22) shows that $\phi^j$ is $1 + \frac{1}{\lambda}m_j 4R^2$-smooth.

Combining the above with the fact that the convex function $\mathrm{dist}(x, \mathcal{X})$ has 1-Lipschitz gradient [6, Proposition 12.30] proves the desired smoothness bound on the function $V_\sigma$.

**LSI Constant bound.** The result follows, for example, by applying the sufficient log-Sobolev inequality criterion stated in [13, Corollary 2.1, Equation (2.3)], combined with the bound (22). The exact constant appearing in the log-Sobolev inequality can be traced from [13, Equation (3.10)].

# F Numerical Experiments

In this section, we numerically validate our main theoretical results presented in Theorems 1 and 2. We empirically demonstrate the necessity of damping Sinkhorn iterations when $\tau < \lambda/2$. In addition,

we examine empirical convergence rates of Algorithms 1 and 2 (for a specific implementation of the approximate Sinkhorn oracle described below) in a simulation setup comprised of isotropic Gaussian measures.[1]

**Simulation setup.** Let $\mu^*_{\lambda,\tau}$ denote the optimal $(\lambda, \tau)$-barycenter; that is, $\mu^*_{\lambda,\tau}$ is the unique solution to the optimization problem (2). Let $(\mu_t)_{t \geq 0}$ denote the iterates of either Algorithm 1 or Algorithm 2. Then, the dual objective sub-optimality gap bounds of Theorems 1 and 2 combined with Lemma 1 establish the convergence of $\text{KL}(\mu^*_{\lambda,\tau}, \mu_t)$ to zero as $t$ goes to infinity.

To numerically compute $\text{KL}(\mu^*_{\lambda,\tau}, \mu_t)$ we need to know the true $(\lambda, \tau)$-barycenter $\mu^*_{\lambda,\tau}$. The only setup where $\mu^*_{\lambda,\tau}$ admits a known closed-form expression is when the marginal measures $\nu^j$ are isotropic Gaussians with identical variance. In particular, it was shown in [17, Proposition 3.4] that for $\nu^j = N(m_j, \sigma^2 I_d)$ and for any non-negative weights $(w_j)_{j=1}^k$ that sum to one, we have

$$\mu^*_{\lambda,\tau} = N\left(\sum_{j=1}^k w_j m_j, \xi^2 I_d\right), \quad \text{where} \quad \xi^2 = \frac{\left(\sigma^2 + \sqrt{(\sigma^2 - \lambda)^2 + 4\sigma^2\tau}\right)^2 - \lambda^2}{4\sigma^2}.$$

Hence, in all the simulations performed in this section, we let $\nu^j = N(m_j, \sigma^2 I_d)$ for some $m_j \in \mathbb{R}^d$ and $\sigma^2 > 0$. While this simulation setup allows for exact computations of the divergence measure $\text{KL}(\mu^*_{\lambda,\tau}, \mu_t)$, there are two primary limitations in our experimental design. First, the theoretical results in this paper are proved under boundedness assumptions on the marginal measures, while Gaussian distributions are unbounded. Second, our simulations do not cover the free-support discrete point clouds setup investigated in Section 4.1 concerning the approximate Sinkhorn oracle implementable via Monte Carlo sampling.

**Implementation of Algorithm 1.** When the marginals $\nu^j$ are Gaussian measures (not necessarily isotropic) and when $\psi_0^j = 0$, then Sinkhorn updates admit a closed-form expression that result in quadratic Sinkhorn potentials $\psi^j, \phi^j$ and Gaussian measure $\mu_t$. In particular, the iterates of Algorithm 1 can be written as

$$\psi_t^j(y) = \frac{1}{2} y^\top A_t^j y - y^\top b_t^j + \text{const}, \quad \phi_t^j(y) = \frac{1}{2} x^\top C_t^j x - x^\top d_t^j + \text{const}, \quad \mu_t = N(e_t, \Sigma_t) \quad (23)$$

for some matrices $A_t, C_t, \Sigma_t \in \mathbb{R}^{d \times d}$ and vectors $b_t, d_t, e_t \in \mathbb{R}^d$ that can be computed using explicit recursive expressions (see, e.g., [34, 41]).

**Implementation of Algorithm 2.** We implement approximate Sinkhorn oracle (Definition 1) as follows. Our updates maintain the property that the potentials $\psi^j$ and $\phi^j$ are quadratic functions of the form described in Equation (23). At every iteration $t$, we replace each matrix $A_t^j$ (for $j = 1, \ldots, k$) by performing the transformation $A_t^j \mapsto A_t^j - N_t^j$, where $N_t^j$ is a random positive-definite matrix with trace equal to $\varepsilon$. In our simulations, each $N_t^j$ is obtained by drawing an independent $d \times d$ matrix $U$ with i.i.d. $N(0, 1)$ entries and setting $N_t^j = \varepsilon \cdot U^\top U / \text{trace}(U^\top U)$. The parameter $\varepsilon$ controls the approximation error of the implemented approximate Sinkhorn oracle.

**Simulation results.** We now comment on the main findings in our numerical simulations.

In Figure 1, we consider the toy setup $k = 1, \nu^1 = N(0, 1)$. Figure 1a shows that undamped iterates explode when $\tau < \lambda/2$ (note the absence of the blue line due to the explosion of iterates when $\lambda/\tau = 2.1$). Figure 1b investigates the critical case $\tau \approx \lambda/2$, noticing a sharp phase transition in the convergence behavior. Figure 1c demonstrates that damping removes the bad behavior. Moreover, the damping factor suggested in our work yields the fastest convergence among the tested five different damping factor choices.

In Figure 2, we investigate an undamped inexact algorithm described above. As suggested by Theorem 2, the iterates converge up to some level governed by the accuracy parameter $\varepsilon$ of the inexact Sinkhorn oracle.

---

[1]The code for reproducing the simulation results is available at `https://github.com/TomasVaskevicius/doubly-entropic-barycenters`.

In Figure 3, we perform the same simulations as in Figure 2, but this time with damping. We observe again that damping helps significantly in the critical case $\tau \approx \lambda/2$, yielding a much faster convergence rate. The second row in Figure 3 considers the overdamped case, where we observe slightly slower convergence than that with the optimal choice of the damping parameter.

In Figure 4, investigate the effect of different levels of damping in the noisy setup (Algorithm 2). First, observe that the undamped algorithm ($\eta = 1$) explodes in the noisy case with $\tau = \lambda/2$, in contrast to the zero-noise setup reported in Figure 1a. Moreover, we find that overdamping might have a negative effect on attained accuracy: larger damping factors lead to lower accuracy at convergence in noisy setups.

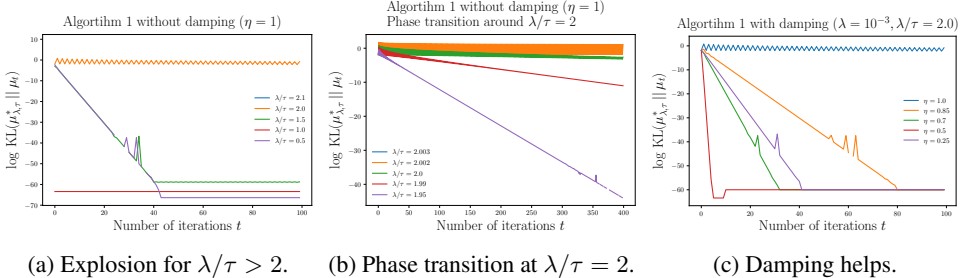

(a) Explosion for $\lambda/\tau > 2$.  (b) Phase transition at $\lambda/\tau = 2$.  (c) Damping helps.

Figure 1: Simulations for Algorithm 1 with $k = 1$, $\nu^1 = N(0, 1)$.

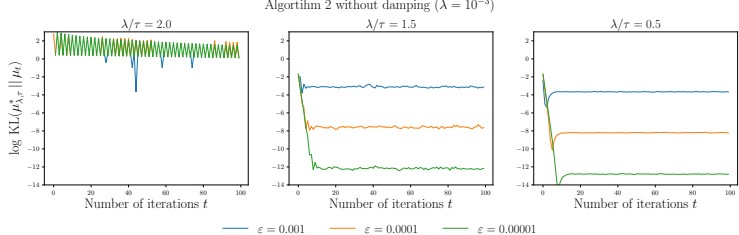

Figure 2: Setup: $k = 3$, $w = (1/3, 1/3, 1/3)$, $\nu^1 = \nu^2 = \nu^3 = N(0, I_{10})$. The effect of noise for the inexact algorithm.

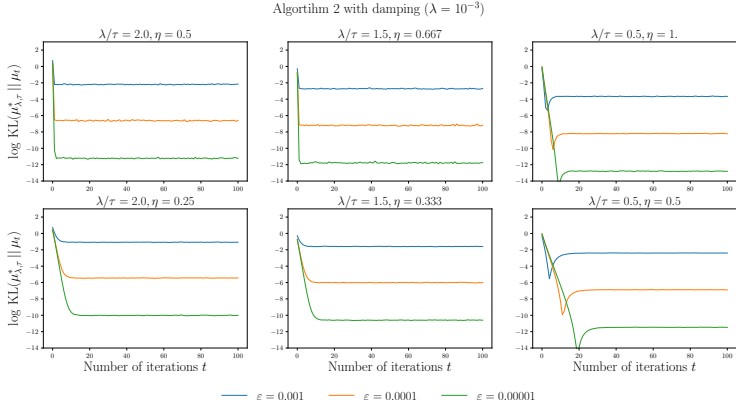

Figure 3: Setup: $k = 3$, $w = (1/3, 1/3, 1/3)$, $\nu^1 = \nu^2 = \nu^3 = N(0, I_{10})$. The effect of noise for the inexact algorithm. Second row overdamps.

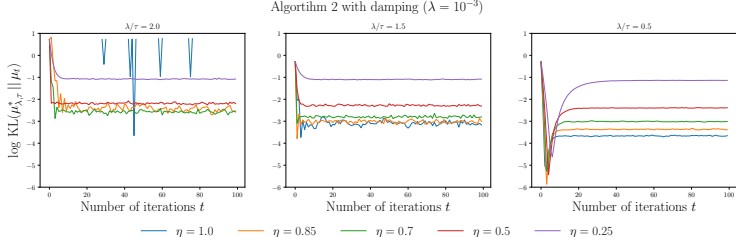

Figure 4: Setup: $k = 3$, $w = (1/3, 1/3, 1/3)$, $\nu^1 = \nu^2 = \nu^3 = N(0, I_{10})$. The effect of damping for the inexact algorithm.

