argument presented below works for any cost function $c$ such that $c(\cdot, y)$ is Lipschitz on $\mathcal{X}$ and grows quadratically at infinity. However, to not cloud the whole picture with technical details, we shall simply take $c(x, y) = \|x - y\|_2^2$. The exact problem setup is formalized below.

**Problem Setting 1.** We consider the setting described in Section 4.1. In addition, suppose that

    1. the reference measure $\pi_{\mathrm{ref}}(dx)$ is the Lebesgue measure (free-support setup);

    2. $\mathcal{X} \subseteq \mathcal{B}_R = \{x \in \mathbb{R}^d : \|x\|_2 \leq R\}$ for some constant $R < \infty$;

    3. $c : \mathbb{R}^d \times \mathbb{R}^d \to [0, \infty)$ is defined by $c(x, y) = \|x - y\|_2^2$;

    4. for any $\boldsymbol{\psi}$ generated by Algorithm 1 we have access to a stationary point $x_{\boldsymbol{\psi}}$ of $V_{\boldsymbol{\psi}}$ over $\mathcal{X}$.

The final condition can be implemented in polynomial time using a first order gradient method. The implication of this condition is that by [55, Lemma 1], for any $\sigma > 0$, the initialization scheme $x_0 \sim \mathcal{N}(x_{\boldsymbol{\psi}}, I_d)$ for the Langevin algorithm (19) satisfies

$$\mathrm{KL}(\rho_0, \mu_{\boldsymbol{\psi},\sigma}) \leq \frac{c_\infty(\mathcal{X})}{\tau} + \frac{d}{2}\log\frac{L_\sigma}{2\pi},$$

where $L_\sigma$ is the smoothness constant of $V_{\boldsymbol{\psi}}/\tau + \mathrm{dist}(x, \mathcal{X})/(2\sigma^2)$ (see Lemma 5).

The following properties are satisfied by the measure $\mu_{\boldsymbol{\psi},\sigma}$.

**Lemma 5.** *Consider the setup described in Problem Setting 1. Let $\psi$ be any iterate generated by Algorithm 2 and let $\mu_{\psi,\sigma}$ be the distribution defined in (20). Then, the measure $\mu_{\psi,\sigma}$ satisfies the following properties:*

*1. For any $\sigma \in (0, 1/4]$ it holds that*

$$\|\mu_\psi - \mu_{\psi,\sigma}\|_{\mathrm{TV}} \le 2\sigma \exp\left(\frac{8R^2}{\tau}\right)\left[\left(4Rd^{-1/4}\right)^{d-1} + 1\right].$$

*2. Let $V_\sigma(x) = \exp(-V_\psi(x)/\tau - \mathrm{dist}(x, \mathcal{X})^2/(2\sigma^2))$; thus $\mu_{\psi,\sigma}(dx) = \exp(-V_\sigma(x))dx$. The function $V_\sigma$ has $L_\sigma$-Lipschitz gradient where*

$$L_\sigma = \frac{1}{\tau} + \frac{1}{\tau\lambda}4R^2 \max_j m_j + \frac{1}{\sigma^2}.$$

*3. The measure $\mu_{\psi,\sigma}$ satisfies LSI with a constant $C_\sigma = \mathrm{poly}(R, \exp(R^2/\tau), L_\sigma)$.*

Above, the notation $C = \mathrm{poly}(x, y, z)$ denotes a constant that depends polynomially on $x, y$ and $z$.

Before proving this lemma, let us state and prove the main result of this section.

**Corollary 1.** *Consider the setup described in Problem Setting 1. Then, for any confidence parameter $\delta \in (0, 1)$ and any accuracy parameter $\varepsilon > 0$, we can simulate a step of Algorithm 2 with success probability at least $1 - \delta$ in time polynomial in*

$$\varepsilon^{-1}, d, R, \exp(R^2/\tau), (Rd^{-1/4})^d, \tau^{-1}, \lambda^{-1}, d, m, \log(m/\delta).$$

Comparing the above guarantee with the discussion at the end of Section 4.1, we see an additional polynomial dependence on $(Rd^{-1/4})^d$. We believe this term to be an artefact of our analysis, which appears due to the total variation norm approximation bound in Lemma 5. Ignoring this term (or considering the setup with $R \le