# OpenReview forum: "Computational Guarantees for Doubly Entropic Wasserstein Barycenters"
_NeurIPS.cc/2023/Conference — NeurIPS 2023 poster_

### Official Review · Reviewer_bjAF · 2023-06-15

**Soundness:** 4 excellent
**Presentation:** 4 excellent
**Contribution:** 4 excellent
**Rating:** 8
**Confidence:** 5

**Summary:**

The paper presents an algorithm (damped Sinkhorn) and theoretical convergence guarantees for computing doubly regularized Wasserstein barycenters. The concept of doubly entropic Wasserstein barycenters extends the single entropic regularized barycenters by introducing an additional level of regularization. This addition allows for the two regularization terms to counterbalance each other, leading to a debiasing effect under the right conditions. The authors demonstrate that the damped Sinkhorn algorithm can be implemented using an approximate Sinkhorn oracle. The latter can be evaluated using Monte Carlo sampling, providing a complete computational pipeline with theoretical guarantees.

**Strengths:**

The paper tackles a complex and highly relevant problem in optimal transport and computational geometry: computing Wasserstein barycenters.

The convergence guarantees are rigorous and well-explained, enhancing the value of the proposed algorithm.

The paper connects the algorithmic problem to practical computation, demonstrating how the proposed methods can be implemented using Monte Carlo sampling. This gives a complete end-to-end view of the problem, which is valuable.


**Weaknesses:**

The paper has some minor errors and inconsistencies, such as missing weights in Equation (2) and inconsistent notations (lack of boldface in $\psi^*$) in Lemma 1 and Theorem 1. There is a typo in line 203 (“is can”). In Definition 1, both $\widehat\nu$ and $\widetilde\nu$ are used.

In the discussion above Lemma 2, the claim about achieving accuracy close to $\sqrt{\varepsilon_\mu}$ is misleading as the bound seems to be larger than $(m_j \varepsilon_\mu)^{1/4}$ based on the first two terms in the bound.


**Questions:**

The LSI constant derived using Holley--Stroock perturbation seems impractical due to its large value. Are there any more practical situations where this constant could be more manageable?

What can be said about the convergence of the algorithm for an unbounded cost, e.g., the quadratic cost over $\mathbb R^d$?


**Limitations:**

Yes.

---

> ### Author Rebuttal · Authors · 2023-08-10
>
> Thank you for your review and for spotting some typos.
>
> - The LSI constant is indeed large, which renders the computational speed exponential in $R/\tau$, where $R$ is the radius of the domain. This is, however, unavoidable because Wasserstein barycenters are NP-hard to compute for discrete point clouds (https://arxiv.org/abs/2101.01100). This means we must pay an exponential cost if we let $\lambda$ and $\tau$ go to zero.
>
> Concerning interesting practical situations where the cost could be manageable, we may consider the following:
>
> 1. Use $\lambda,\tau = \Theta(1)$; this way, we lose approximation properties but gain in statistical properties and in computational speed;
>
> 2. Work with specific classes of marginals, e.g., it is possible to carry out exact computations for Gaussian marginals, where we can actually implement the exact scheme (Algorithm 1);
>
> 3. There is some hope that some efficiency could be gained, for example, for log-concave measures, again relying on sampling approaches. However, we are not aware of any specific results in this direction yet.
>
>
> - Regarding convergence in the unbounded case, we believe this should be possible. It is known how to analyze Sinkhorn's algorithm in the unbounded case (see https://arxiv.org/abs/2212.06000), and because our scheme essentially combines Sinkhorn updates with damping, we believe it should not be too difficult to extend our results to quadratic cost in $\mathbb{R}^{d}$.

---

> > ### Comment · Reviewer_bjAF · 2023-08-11
> >
> > Thank you for your response, I am satsfied and I will keep my score. Unlike the other reviewers, I did not find the lack of experiments to be a detriment to the paper. Given the general lack of theory for computing barycenters (as mentioned, this seems to be the first convergence guarantee of its kind for the setting under consideration) it is clear that the theoretical result is already considerably interesting. Therefore, I strongly advocate for acceptance.

---

### Official Review · Reviewer_ofnZ · 2023-06-29

**Soundness:** 4 excellent
**Presentation:** 4 excellent
**Contribution:** 2 fair
**Rating:** 8
**Confidence:** 5

**Summary:**

-----
EDIT : 6 --> 8
-----


This paper builds on [15] and considers the recently introduced model of _doubly regularized entropic Wasserstein Barycenters_ which, given a set of measures $\nu^1,\dots, \nu^k$, weights $(w_j)_j$ (non-negative and sum to $1$), a reference measure $\pi$, and two smoothing parameters $\lambda,\tau > 0$, consider the problem

$$ \text{ minimize } \mu \mapsto \tau \mathrm{KL}(\mu,\pi) + \sum_{j=1}^k w_j T_\lambda(\mu, \nu_j), $$

where

$$ T_\lambda(\mu,\nu) = \inf_\gamma \braket{c,\gamma} + \lambda \mathrm{KL}(\gamma,\mu\otimes\nu). $$

Depending on the choice of the parameters $\lambda,\tau$, this problem encompasses many standard formulations for Wasserstein barycenters and their variations (Schrödinger barycenters, etc.).

This problem is deeply analyzed in [15] from a theoretical perspective (existence, uniqueness, smoothness of solutions, etc.), which concludes on the need to elaborate on the computational complexity of these $(\lambda,\tau)$-barycenters. This is the purpose of the current paper, which has two (three) main contributions to that respect :
1. They provide a theoretical _damped Sinkhorn algorithm_ (which can been seen as an adaptation of _Iterative Bregman Projections_ of [6]) for which they prove a $O(t^{-1})$ convergence rate toward the global minimizer of the above functional ($t$ being the number of steps). In a nutshell, the trick consists of replacing the "learning rate" $\lambda$ in the standard Sinkhorn algorithm by the quantity $\min(\lambda,\tau)$.
2. They provide a  _approximated_ algorithm for which an implementation is possible, provided one has access to an "ApproximatedSinkhornOracle", and for which convergence rate are accessible,
3. (2bis) They provide an example of such a practical Approximated Sinkhorn Oracle based on random sampling techniques.


### References
[15] : (Doubly regularized Entropic Wasserstein Barycenters, Chizat, 2023)

**Strengths:**

First, the paper is overall very well written. It manages to remain concise without sacrificing the mathematical rigor, and basically every line/equation is interesting, which is highly appreciated.

I also believe that its theoretical contributions are fairly significant and quite insightful and clearly speak in favor of adding more _outer regularization_ in OT-related problems.

The proofs have been investigated (with the exception of those of Lemma 2 and 3) and no major flaw was detected; the clarity of the presentation make them easy to parse while not being trivial.

**Weaknesses:**

# Major issue

**The paper feels somewhat incomplete.** This is my main (and almost single) issue with the work (which undoubtedly has great potential). In a nutshell, this work is dedicated to the elaboration of provably converging **practical** algorithms to compute $(\lambda,\tau)$-barycenters; hence having absolutely no numerical illustration of such algorithms is somewhat surprising---even more given that the implementation seems to exist ($\ell$199, "we have observed empirically"; why not showcasing it?).

Typically, the divergence of the standard Sinkhorn algorithm (when $\tau < \lambda/2$) and the convergence of the damped one would have been nice to observe (especially given that the former is only empirical). Similarly, the paper mentions both in the abstract and in the conclusion that the approach works for _both fixed and free-support_ models (which boils down to the choice of the reference measure $\pi$), but this is (almost) not developed in the work. In my opinion, it is fascinating to have a globally converging way to approximate Wasserstein barycenter, and showcasing empirical situations where the proposed approach provides much better results than the "naive" one (say, IBP).

Similarly, the work mentions at several occasions the edge case $\tau = \lambda/2$ (because of the powerful statistical properties showcased in [15]), but never leverage this setting as far as I can tell. I do not know if there is something specific to say from the algorithmic perspective, but here as well, I feel like this could have been a nice occasion to numerically showcase the debiasing effect (for instance).

Do not get me wrong: I believe that the current contributions of the paper are of interest. But in my opinion, there are few elements that are missing to make the paper complete.

# Minor issues and other remark/suggestions

Note: this are not actual criticism that are expected to be addressed during the discussion period immediatly, but rather comments that may hopefully be useful for the authors.

- The hyperlinks are broken (probably due to the split of the pdf between the main body and the supplementary material). It may be nice to fix that for the camera ready version (which implies, I guess, to compile the supplementary material independently. Note that you can include the main body in the supplementary, which makes it "self contained" and quite handy for reviewers/readers).
- [typo] In Eq. (2), I think that the $(w_j)_j$ are missing. This occurs in other places as well (Eq 8, def of $\mu_\psi, Z_\phi$).
- [typo] The parameters $\nu,w, \lambda,\tau$ for the objective function $E$ are sometime missing (e.g. in Eq 10). This can be slightly confusing given the proximity with the expectation operator $\mathbf{E}$.
- [typo] $\ell$193, $\nu_j^t$ should be $\nu_t^j$ I guess.
-  [typo] In definition 1, the notation $\hat{\nu}$ and $\tilde{\nu}$ are both used (in my understanding, they denote the same quantity).
- [typo] $\ell$232, I think it should be "the proof _of its convergence_ is deferred...".
- [suggestion] That's trivial but it may be handy to write in $\ell$215 that  $\delta_{t}-\delta_{t+1} = E_{\lambda,\tau}^{\nu,w} (\phi_{t+1}) - E_{\lambda,\tau}^{\nu,w} (\phi_{t}) \geq 0$ by Prop 1, making the sequence of error non-increasing (instead of saying it in $\ell$222).
- [suggestion] Lemma 1 is never referenced after being stated (only once in the contributions section). While it is of course obvious, it may be worth adding a short sentence after Theorem 1 saying something like "Theorem 1 together with Lemma 1 implies the convergence of $\mu_t$ toward $\mu_{\lambda,\tau}$.
- [typo] $\ell$430, I think that $\Delta_t^j$ should be defined without the $\log$.

**Questions:**

# Main questions

These are questions/suggestion I would like to see addressed during the discussion period. Given what I have written above, they naturally rely on (synthetic) numerical experiments that may hopefully shed some light on the strengths and weaknesses of the proposed approach.

1. Is there a computational price to pay for the _damped_ Sinkhorn? Namely, from what is written in the paper, I assume that the "naive" Sinkhorn algorithm (with $\eta = 1$ no matter $\tau$) does converge when $\tau \in [\lambda/2 , \lambda]$ (at least empirically). If so, (i) does it provide (empirically) the same output (asymptotically in $t$) as the output of the damped algorithm? (ii) if yes, is it faster? (I would expect that it is, since the damped algorithm somewhat slow the gradient descent).
2. Can you elaborate on $\pi_{\mathrm{ref}}$ be the Lebesgue measure encode "the free support case"? If my understanding is correct, this forces $\mu$ to have a density, which is not exactly what "free-support" refers to in the work of Cuturi and Doucet (2014) (but I agree that their terminology can be discussed as well as they fix the cardinality of the support). But typically, if the $\nu^j$ are discrete, their $(0,0)$-barycenter is discrete as well (Carlier & Ekeland, 2015) and in that case, it may be hard for $\mu_t$ to concentrate. From [15, Thm 3.2] we may expect a $\lambda^2$ proximity between the exact Wasserstein barycenter and the $(\lambda,\lambda/2)$ one, but as far as I can tell this theorem is proved only in the case where the measures have densities. So I would be interested to see if/how much things fail in the discrete setting (now that it is possible to implement it!).


# Minor questions

These are minor questions that I'm asking because I am interested in the work, but I would not be offended if the authors do not address them during the discussion period.

- From my reading of the proofs, I missed the point where using the damped algorithm is crucial. I think that I have a correct line-by-line understanding of the proof, but not a global vision; could you tell me what would fail if one tries to run the same proof but with $\eta = 1$? I guess it appears in the proof of Lemma 4 (case $\tau < \lambda$), which I did not investigated; is it possible to have a global picture of why it is required?
- In the way Theorem 2 is currently stated, nothing is said once the condition on $T$ is reached (assuming $T < \infty$). Is it clear that once the condition is reached, the gap remains below $2\epsilon$? As far as I can tell by checking the proof, it is not sure that the sequence $\tilde{\delta}(t) - \tilde{\delta}(t+1)$ is non-increasing (once the criterion is reached), may it happen that the gap increases again in an uncontrolled way? (i.e. at time $T$ we are below $2\epsilon$, but at $T+1$ we suddenly get something much worse and we have to keep running the gradient descent again) I think that (if such thing can happen) this can be avoided by simply checking the variations of the objective value.

**Limitations:**

I do not identify specific limitation of potential negative societal impact specific to this work.

For the former point, I guess that additional numerical experiments may showcase some limitations of the work, that would be worth discussing then (without diminishing the contributions of the work).

---

> ### Author Rebuttal · Authors · 2023-08-10
>
> Thank you for such a thorough review and for your very helpful suggestions.
>
> ## Answers to Main Questions
>
> ### Question 1: Computational Price of Damping
>
> *Is there a computational price to pay for the damped Sinkhorn?*
>
> **Answer:** Actually, in the numerical experiments attached to the main response, we have observed that the correct amount of damping helps to speed up the convergence speed. Indeed, Figure 1 (c) shows that $\eta = 0.5$ gives the fastest convergence (which, in this setup, is the choice suggested in the paper). Also, you may inspect Figures 2 and 3. We have three rows: no damping, theoretically suggested damping and overdamping. In all three columns, we see that the middle row (theoretical choice) gives the fastest convergence.
>
> However, there is a second side to this. In particular, Figure 4 suggests that damping more is worse in noisy setups. In addition, by overdamping beyond the theoretical choice specified in Algorithm 1, we do see a slowdown as expected (see Figure 3, compare the top row with the bottom one).
>
> *Namely, from what is written in the paper, I assume that the "naive" Sinkhorn algorithm does converge when $\tau \in [\lambda/2, \lambda]$ (at least empirically).*
>
> **Answer:** This is true, which can now be seen in Figure 1 (c). Indeed, it also answers your question (i) that we asymptotically converge to the same value regardless of the choice of $\eta \in (0,1]$, and we have already discussed (ii) above.
>
> ### Question 2: On the Reference Measure
>
> You are absolutely right that whenever $\pi\_{\mathrm{ref}}$ has a density, it forces the optimal $(\lambda, \tau)$-barycenter also to have a density. And for arbitrary discrete marginals, with $\pi\_{\mathrm{ref}}$ being the Lebesgue measure, the convergence via our Algorithm 2 would indeed be slow as $\tau$ goes to $0$ (our convergence guarantee is exponential in $R/\tau$ where $R$ is the radius of the domain). However, this is unavoidable due to the NP-hardness results for approximating $(0,0)$ barycenters due to https://arxiv.org/abs/2101.01100.
>
> It is true that the $\lambda^2$ proximity to the true $(0,0)$-barycenter only holds in the continuous and smooth case. However, we can benefit from the debiaising effect of the $(\lambda,\lambda/2)$ barycenters -- even when we are doing computations with discrete measures -- when there is an underlying continuous structure. Consider for instance the common setting where the discrete marginals are obtained by discretizing (say, via $n$ iid samples) some continuous distributions and we would like to compute/estimate the unregularized barycenter of the continuous distributions directly. Combining the approximation ($\lambda^2$) and estimation ($\lambda^{-1-d/2}n^{-1/2}$) errors leads to better estimation bounds using $(\lambda,\lambda/2)$-barycenters compared to other entropy regularized barycenters (see [15]); even though the computation is done with discrete marginals.
>
>
> ## Answers to Minor Questions
>
> - Here, I hope the performed numerical simulations will be helpful. Consider the $\tau = \lambda/2$ cases shown in Figure 1 (a) and Figure 2 (left-most plot). Both plots display a zig-zag-type line for the undamped algorithm ($\eta = 1$), which essentially shows a behavior analogous to the one encountered in quadratic optimization for gradient descent with too large step size, which is when gradient descent iterates bounce back and forth between different sides of a valley. In the gradient descent case, this zig-zag behavior arises for smooth problems with a too-large step size, which is also what is happening in our simulations here. Indeed, there is a precise sense in which smoothness of the dual maximization objective for doubly entropic barycenters can be characterized [Proposition 2.6., https://arxiv.org/pdf/2303.11844.pdf], which shows that decreasing $\tau$ (unsurprisingly) makes the dual objective less smooth, which is detrimental to the $\eta=1$ algorithm as its convergence is essentially smoothness based. What matters is the $\tau$ relation to $\lambda$, and as $\tau$ passes below $\lambda/2$, we move beyond a critical threshold at which the exact alternating maximization/minimization stops working (see Figure 1 (b) for a display of this phase transition).
>
> - You are absolutely right. We do not say anything about what happens after the condition is reached. Indeed, numerical simulations (right-most column in Figures 2,3,4) demonstrate a curious behavior, where Algorithm 2 first reaches a good value and later gets stuck into a slightly worse one, where it remains indefinitely.

---

> > ### Comment · Reviewer_ofnZ · 2023-08-12
> > **Thanks!**
> >
> > Thank you for your detailed answer and the additional pdf including few experiments (that are worth including in the main paper if eventually accepted).
> > I am quite convinced by the depth of the work and will increase my grade.

---

### Official Review · Reviewer_9mYE · 2023-07-04

**Soundness:** 4 excellent
**Presentation:** 3 good
**Contribution:** 2 fair
**Rating:** 4
**Confidence:** 4

**Summary:**

The paper has proposed a computational algorithm for computing the newly developed regularized Wasserstein barycenters in [Chizat,2023] via optimizing the duality of the primal problem. The paper also characterised the convergence of algorithms on both exact and approximated algorithms which are the main contributions of the paper.

**Strengths:**

The algorithms and their convergence guarantee for computing doubly entropic Wasserstein barycenters are significant contributions in terms of technical aspects. The paper is easy to follow as the related results have been well presented to make use of the derivations of the proposed algorithms.

**Weaknesses:**

There are some aspects the paper can be improved:
- A section for numerical experiments/illustrations for both synthetic and real-world data will help to demonstrate the results of the two main algorithms and their convergence algorithms.
- Algorithms should link to the text updating equations. For instance, in Alg. 1, step 2(a) computes the left part of the equation (11) while step 2(b-d) computes the right part of that equation, authors break down into multiple lines of code without clear purposes which usually make readers harder to follow.
- In line 199, "We have observed empirically that the iterates of the iterative Bregman projections (i.e., the scheme of updates (12), (11)) diverge whenever \tau < \lamba<=2", it is not clear where the results with the iterative Bregman projections presented.

**Questions:**

 As the doubly entropic Wasserstein barycenter may have some interesting properties for some specific type of data if the author can discuss further in terms of theoretical and experimental aspects that would be great.

**Limitations:**

No empirical result provided.

---

> ### Author Rebuttal · Authors · 2023-08-10
>
> Thank you for the review.
>
> The theoretical aspects of doubly entropic barycenters have been thoroughly investigated in https://arxiv.org/pdf/2303.11844.pdf. Our primary goal was, instead, to provide new numerical schemes for their computation and to establish their convergence, particularly covering the case $\tau < \lambda$ for which we obtained the first provably convergent method.
>
> In addition, some numerical examples concerning a comparison of $(\lambda, \tau)$-barycenters with different parameter values are available in Section 6 of the referenced paper.

---

### Official Review · Reviewer_uehn · 2023-07-06

**Soundness:** 3 good
**Presentation:** 4 excellent
**Contribution:** 3 good
**Rating:** 7
**Confidence:** 4

**Summary:**

This paper proposes an algorithm for solving the doubly regularized Wasserstein barycenter problem for probability measures that corresponds to adding an inner regularization based on the entropy penalty appearing in the Wasserstein distance term, and an outer regularization appearing at the level of the Wasserstein barycenter problem, based on the KL divergence. The algorithm comes with convergence guarantees, depending on inner and outer regularization parameters.The algorithm is then modified to approximately solve the doubly regularized Wasserstein barycenter problem, but with non-asymptotic convergence guarantees.

**Strengths:**

This article is very well written, proposing an algorithm for a recently introduced notion of regularization for the Wasserstein barycenter, unifying several proposals in the literature. The proposed algorithm does not rely on space discretization (prohibitive in high dimensions). The proofs are accurate, well documented and complete. The most important contribution lies in the approximate damped Sinkhorn scheme, which allows for non-asymptotic convergence guarantees on the value of the dual objective.

**Weaknesses:**

The idea of dumping an algorithm is not really innovative, but it stabilizes the algorithm and provides convergence guarantees for any positive values of the parameters $(\lambda,\tau)$.

A result on the distance between barycenters obtained via the exact and approximate damped Sinkhorn scheme could have been interesting. In particular, the addition of experiments on the barycenters obtained via both algorithms would strengthen the paper.

Finally, an intuition on the approximate Sinkhorn oracle (Definition 1) is important in my opinion, as the properties listed in Definition 1 are quite precise. An example of a Radon-Nikodym derivative that does not satisfy these properties could be added.

**Questions:**

Probability measures are supported on a compact convex subset of $\mathbb{R}^d$, could this be relaxed?

In practice, how does the number of support points of the discrete barycenter is chosen? More generally, how do you choose $\pi_{ref}$?

**Limitations:**

An experimental (and if possible theoretical) comparison of the barycenters obtained by the exact and approximate proposed algorithms, as well as a comparison with Bregman iterative projections algorithm for particular choices of regularization parameters would, in my opinion, improve this article.

---

> ### Author Rebuttal · Authors · 2023-08-10
>
> Thank you for your comments and questions.
>
> - Regarding the compactness assumption, it is really only needed in our case because the prior work that introduced doubly entropic barycenters derived many theoretical results in the compact case. In particular, compactness was used in that context to justify a certain interchange of inf-sup [Section 2.3, https://arxiv.org/pdf/2303.11844.pdf]. Because these types of results can usually be obtained via other means (e.g., by approximation), and because it is known how to analyze Sinkhorn's algorithm for unbounded measures, we do not see any major obstacles in extending our results beyond the compact case.
>
> - Regarding the choice of $\pi\_{\mathrm{ref}}$, there is, in general, no way to know where the optimal barycenter is supported or in which part of the region it is concentrated without computing it first. Thus, you would typically choose $\pi\_{\mathrm{ref}}$ by gridding the space. Of course, in high dimensions, this approach is not feasible. This motivates looking into grid-free/support-free methods such as Algorithm 2 with the implementation suggested following Lemma 2.

---

> > ### Comment · Reviewer_uehn · 2023-08-16
> > **Response to the authors**
> >
> > Thank you for your response and for producing an additional section of convincing experiments. I have also read the responses of the other reviewers, who have shed light on the contribution of your work, and have decided to increase my score from 5 to 7.

---

### Official Review · Reviewer_nAci · 2023-07-07

**Soundness:** 3 good
**Presentation:** 3 good
**Contribution:** 3 good
**Rating:** 7
**Confidence:** 3

**Summary:**

This paper presents a study on the computation of doubly regularized Wasserstein barycenters, a recently introduced family of entropic barycenters with inner and outer regularization strengths. The authors build upon previous research, which has shown that different choices of regularization parameters unify various entropy-penalized barycenter concepts, including debiased barycenters.

The proposed algorithm for computing doubly regularized Wasserstein barycenters combines damped Sinkhorn iterations with exact maximization/minimization steps, ensuring convergence for any choice of regularization parameters. Additionally, the authors introduce an inexact variant of the algorithm that utilizes Sinkhorn Oracle definition, providing non-asymptotic convergence guarantees for approximating Wasserstein barycenters between discrete point clouds in the free-support/grid-free setting. The authors highlight that while a straightforward adaptation of the alternate maximization scheme leads to diverging iterates for small values of $\tau$, their analysis demonstrates that damping these iterations is sufficient to achieve convergence.


**Strengths:**

1. Clear structure, good mathematical exposition, and solid theoretical results.
2. Detailed proofs to support their claims, demonstrating a strong theoretical foundation


**Weaknesses:**

1. Lack of numerical experiments and empirical evaluations.
2. While the paper discusses computational complexity, further analysis and discussion on the scalability and efficiency of the algorithm could provide a more comprehensive understanding of its practical implications.


**Questions:**

I have the following questions/comments for the authors:

1. Can the authors provide more motivation for using the definition of Sinkhorn Oracle in Section 4, Approximate Damped Sinkhorn Scheme? It would be helpful to understand the rationale behind this choice and how it contributes to the proposed algorithm.
2. In Section 2.2, Doubly Regularized Entropic Barycenters, could the optimization part (Line 172 - 184) be made more clear and detailed? This section seems to be the foundation for the updates in your main contribution algorithm. Providing additional explanations and elaboration would enhance the understanding of the optimization process.
3. Algorithm 1 is presented without a thorough mathematical derivation. It would be beneficial to include a more detailed derivation or explanation of the algorithm to aid readers in understanding the underlying principles and steps involved.


**Limitations:**

While the paper provides theoretical guarantees and analyses, incorporating visualizations and benchmarking against alternative approaches would enhance the practical significance of the proposed algorithm

---

> ### Author Rebuttal · Authors · 2023-08-10
>
> Thank you for your comments and questions. We respond to your questions below:
>
> 1. Let us explain why the Approximate Sinkhorn Oracle (Definition 1) is defined the way it is. First, why do we need an approximate algorithm at all? We need it because, for continuous measures, we typically cannot implement the exact algorithm (Algorithm 1) due to the high-dimensional integrals that need to be computed in lines (a) and (e) of Algorithm 1. Of course, when the marginals are discrete, for any given x we can actually evaluate the line (a), as the integral inside is just a sum. Thus, the only issue is with the computation of line (e), repeated below for convenience:
>
> $$
> (e) \quad \frac{d\nu\_{t}^{j}}{d\nu^{j}}(y) \leftarrow \int \exp\left(\frac{\phi_{t}^{j}(x) + \psi_{t}^{j}(y) - c(x,y)}{\lambda}\right)\mu\_{t}(dx).
> $$
>
> While we cannot compute the integral above exactly, if we can generate samples from $\mu_{t}$, then for any given $y$ we could estimate the desired quantity $\frac{d\nu\_{t}^{j}}{d\nu^{j}}(y)$ up to some error.
>
> This leads to the question: suppose, instead of $\frac{d\nu_{t}^{j}}{d\nu^{j}}(y)$, we have some other function $g^{j}\_{t}(y)$. How close does it need to be to $\frac{d\nu_{t}^{j}}{d\nu^{j}}(y)$ for us to run Algorithm 1 with $g^{j}\_{t}$ instead of $\frac{d\nu_{t}^{j}}{d\nu^{j}}(y)$? Well, the answer to this is laid out in Definition 1. To get these properties, you just need to follow the proof of Theorem 1 and put $g^{j}_{t}$ in place of  $\frac{d\nu\_{t}^{j}}{d\nu^{j}}(y)$, observing what properties need to hold for you to preserve the convergence analysis up to some tolerance. You may then see how the proof of Theorem 2 follows the lines of Theorem 1 when the properties of Definition 1 are plugged into the proof.
>
> We hope this helps!
>
> 2. and 3. Thank you for the suggestions. We will make sure to clarify these parts when we update the paper.

---

> > ### Comment · Reviewer_nAci · 2023-08-17
> >
> > I appreciate the author's detailed response. It addresses most of my concerns.  I'll raise my score accordingly.

---

### Author Rebuttal · Authors · 2023-08-10

We thank all the reviewers for their feedback. We will answer minor questions raised by the reviewers individually; in this shared response to all reviewers, we will focus on the numerical simulations aspect.

Most of the reviewers pointed out the absence of numerical simulations as a significant shortcoming of our paper. We have performed numerical simulations that cover a significant fraction of the concerns raised.

## Note: there is currently no performance benchmark for our algorithm.

Several reviewers suggested benchmarking our algorithm against others. We would like to clarify that:
- We have provided the first provably convergent method for the case $\tau < \lambda$ (both exact and inexact versions).
- Our focus on $\tau < \lambda$ is deliberate. Specifically, the recently discovered case $(\lambda, \lambda/2)$ enjoys very strong theoretical properties, and we have provided the first algorithm covering this case with convergence rates.
- When $\tau \geq \lambda$ (no damping), our algorithm reduces to the classical alternate maximization/minimization scheme, which is well understood from both practical and theoretical perspectives. However, note that we provided a unified analysis that covers all choices of $(\lambda, \tau)$.

## Simulation setup: isotropic Gaussian marginals.

However, even without a benchmark, we can perform insightful simulations to address several issues highlighted by the reviewers. These include:
- demonstrating the need for damping when $\tau < \lambda/2$;
- examining empirical convergence rates with and without damping;
- testing the inexact damped Sinkhorn scheme (Algorithm 2).

To provide meaningful computations, we need a setup for which we know what the true $(\lambda, \tau)$ barycenter is. In this response and in the attached simulations, we denote the optimal barycenter for a given problem by $\mu^{*}\_{\lambda, \tau}$, while the iterates of our algorithms (Algorithm 1 and Algorithm 2) are denoted by $\mu\_{t}$.

The only known case with closed-form $\mu^{*}\_{\lambda, \tau}$ is that of isotropic Gaussian marginals with the same variance (Proposition 3.4 in https://arxiv.org/pdf/2303.11844.pdf). Let $w \in \mathbb{R}^{k}$ denote the non-negative weights vector that sums to one, let $\sigma^{2} > 0$ be arbitrary, and for $j=1,\dots,k$ let $m\_{j} \in \mathbb{R}^{d}$ be arbitrary. If $\nu^{j} = N(m\_{j}, \sigma^{2}I\_{d})$ for $j=1,\dots,k$, then

$$
  \mu^{*}\_{\lambda, \tau} = N(\sum\_{j=1}^{k} w\_{j}m\_{j}, \xi^{2} I\_{d})
  \text{ with } \xi^{2} = \frac{(\sigma^{2} + \sqrt{(\sigma^{2} - \lambda)^{2} + 4\sigma^{2}\tau})^{2} - \lambda^{2}}{4\sigma^{2}}.
$$

## Implementing Algorithm 1.

When the marginals are Gaussian (not necessarily isotropic) and when we initialize $\psi^{j} = 0$, then, there exist some $d\times d$ matrices $A_{t}^{j}, C_{t}^{j}, \Sigma_{t}$ and $d$-dimensional vectors $b_{t}, d_{t}, e_{t}$ such that for all $t$ the iterates of Algorithm 1 satisfy:

$$
  \psi^{j}\_{t}(y) = \frac{1}{2}y^{\top}A\_{t}^{j}y - y^{\top}b\_{t}^{j} + \mathrm{const},\quad
 \phi^{j}\_{t}(y) = \frac{1}{2}x^{\top}C\_{t}^{j}x - x^{\top}d\_{t}^{j} + \mathrm{const},\quad
\mu_{t} = N(e_{t}, \Sigma_{t}).
$$

That is, the functions $\psi$ and $\phi$ are always quadratic, and $\mu_{t}$ is always Gaussian. In this case, all the integrals involved in Algorithm 1 have closed-form solutions, and we can implement Algorithm 1. Remark: for work in this spirit, see https://arxiv.org/abs/2006.02572.

## Implementing Algorithm 2.

We implement Algorithm 2 via the transform $A_{t}^{j} \mapsto A\_{t}^{j} - N\_{t}^{j}$, where $N\_{t}^{j}$ is a random positive semi-definite matrix with trace equal to $\varepsilon$ (see attached simulations). This serves as a toy model for an inexact Sinkhorn oracle for Gaussian simulations (but note that it is different from the Monte Carlo scheme that we proposed for discrete measures).

## Limitations of our experimental design.

Let us briefly summarize some limitations of our simulation setup:
- The results in our paper are proved under boundedness assumptions, while Gaussians are unbounded. While it requires justification, extending our results to unbounded cases should not cause any major problems.
- Our simulations do not cover the free-support discrete point clouds setup.

## Comments on the uploaded simulation results.

We now comment on our simulations, referring to the figures in the attached pdf:
- Figure 1 considers the toy setup $k=1$, $\nu^{1} = N(0,1)$. We observe:
    1. figure (a) shows that undamped iterates explode for $\tau < \lambda/2$ (note the absence of a blue line, because its iterates exploded);
    2. figure (b) zooms in on the case $\tau \approx \lambda/2$, noticing a sharp phase transition in the convergence behavior at the critical case $\tau = \lambda/2$;
    3. figure (c) demonstrates that damping removes the bad behavior. Moreover, the damping factor suggested in our work yields the fastest convergence among the tested five different damping factor $\eta$ choices.
- Figure 2 investigates an undamped inexact algorithm. As suggested by Theorem 2, the iterates converge up to some level governed by the accuracy parameter of the inexact Sinkhorn oracle.
- Figure 3 performs the same simulations, but this time with damping. We observe again that damping helps significantly in the critical case $\lambda/\tau = 2$, yielding a much faster convergence. The second row in Figure 3 considers the overdamped case, where we observe slightly slower convergence than that with the optimal choice of damping parameter.
- Figure 4 investigates the effect of damping in the noisy setup (Algorithm 2). First, observe that the undamped algorithm (eta = 1) explodes in the noisy case with $\tau = \lambda/2$, while this was not the case in the zero-noise setup (Figure 1 (a)). Moreover, we find that overdamping might have a negative effect on attained accuracy: larger damping factors lead to lower accuracy at convergence in noisy setups.

---

### Decision · Program_Chairs · 2023-09-21

**Decision:**

Accept (poster)

**Comment:**

All reviewers agree this work should appear at NeurIPS and will have a welcome audience, so this is a clear accept.  In revising the paper, please implement changes promised in the rebuttal and incorporate text from the reviewer responses.  The AC also agrees with the reviewers who suggest that additional computational experiments in this work would bolster its impact and demonstrate value to practitioners.